# Development of a continuous UAV-mounted air sampler and application to the quantification of $CO_2$ and $CH_4$ emissions from a major coking plant

Tianran Han[1], Conghui Xie[1], Yayong Liu[1], Yanrong Yang[1], Yuheng Zhang[1], Yufei Huang[1], Xiangyu Gao[2], Xiaohua Zhang[3], Fangmin Bao[4], Shao-Meng Li[1]

[1] State Key Joint Laboratory of Environmental Simulation and Pollution Control, College of Environmental Sciences and Engineering, Peking University, Beijing 100871, P.R. China

[2] Beijing Wisdominc Technology Co., Ltd., Beijing, P.R. China

[3] Suzhou Environmental Monitoring Center, Jiangsu Province, P.R. China

[4] Jiangsu Shagang Group Co., Ltd., Beijing, P.R. China

*Correspondence to*: Shao-Meng Li (shaomeng.li@pku.edu.cn)

**Abstract.** The development in uncrewed aerial vehicle (UAV) technologies over the past decade has led to a plethora of platforms that can potentially enable greenhouse gas emission quantification. Here, we report the development of a new air sampler, consisting of a pumped stainless coiled tube of 150 m in length with controlled time-stamping, and its deployment from an industrial UAV to quantify $CO_2$ and $CH_4$ emissions from the main coking plant stacks of a major steel maker in eastern China. Laboratory tests show that the time series of $CO_2$ and $CH_4$ measured using the sampling system is smoothed when compared to online measurement by the cavity ring-down spectrometer (CRDS) analyzer. Further analyses show that the smoothing is akin to a convolution of the true time series signals with a heavy-tailed digital filter. For field test, the air sampler was mounted on the UAV and flown virtual boxes around two stacks in the coking plant at Shagang Steel Group. Mixing ratios of $CO_2$ and $CH_4$ in air and meteorological parameters were measured from the UAV during the test flight. A mass-balance computational algorithm was used on the data to estimate the $CO_2$ and $CH_4$ emission rates from the stacks. Using this algorithm, the emission rates for the two stacks from the coking plant were calculated to be $0.12 \pm 0.014$ t h$^{-1}$ for $CH_4$ and $110 \pm 18$ t h$^{-1}$ for $CO_2$, the latter being in excellent agreement with material balance based estimates. A Gaussian plume inversion approach was also used to derive the emission rates and the results were compared with those derived using the mass-balance algorithm, showing a good agreement between the two methods.

## 1 Introduction

Atmospheric carbon dioxide ($CO_2$) and methane ($CH_4$) are the two major anthropogenic greenhouse gases (GHGs). Both $CO_2$ and $CH_4$ in the atmosphere have been increasing since the industrial revolution, particularly rapidly over the past ten years. Global networks consistently show that the globally averaged annual mean $CO_2$ molar fraction in the atmosphere increased by 5.0% from 2011 to 2019, reaching $409.9 \pm 0.4$ ppm in 2019. Likewise, the globally averaged surface atmospheric molar fraction of $CH_4$ in 2019 was $1866.3 \pm 3.3$ ppb, 3.5% higher than in 2011 (IPCC, 2021). $CH_4$ is a stronger absorber of Earth's thermal infrared radiation than $CO_2$, with its global warming potential (GWP) 32 times greater than that of $CO_2$ over a 100-year horizon (Saunois et al., 2020). Although its molar fractions in the atmosphere are about 200 times lower than those of $CO_2$, the total radiative forcing of $\sim 1.0$ W m$^{-2}$ for $CH_4$ is about half of that of $CO_2$ ($\sim 2$ W m$^{-2}$) (IPCC, 2021), contributed by its direct radiative forcing of $(0.6 \pm 0.1)$ W m$^{-2}$ and indirect forcing of 0.4 W m$^{-2}$ resulting from chemical reactions producing other GHGs including $CO_2$, $O_3$, and stratospheric water (Turner et al., 2019). Furthermore, although global anthropogenic $CH_4$ emissions are estimated to be only 3% of the global anthropogenic $CO_2$ emissions in units of carbon mass flux, the increase in atmospheric $CH_4$ is responsible for about 20% of the warming induced by long-lived greenhouse gases since pre-industrial times (Etminan et al., 2016). Both $CO_2$ and $CH_4$ are produced and released into the atmosphere from a variety of natural and anthropogenic sources. Natural emission sources include vegetation, oceans, volcanoes and naturally occurring wildfires, but most of the increases in atmospheric $CO_2$ and $CH_4$ are considered to have resulted from anthropogenic emissions, from sources including fossil fuel production and uses, agricultural activities, land use and industrial processes (IPCC, 2021).

Quantification of $CO_2$ and $CH_4$ emissions from sources requires continuous measurements of their mixing ratios as well as meteorological parameters using a variety of stationary and mobile platforms, including ground-based vehicles (Rella et al., 2015; Brantley et al., 2014), towers (Helfter et al., 2016; Takano and Ueyama, 2021), aircrafts (Li et al., 2017; Liggio et al., 2019) and sattelites (Miller et al., 2013; Turner et al., 2015). Small uncrewed aerial vehicles (UAVs) have become emerging platforms due to the recent rapid technological developments. They are flexible, versatile and relatively inexpensive. Most importantly, a UAV platform fills the sampling space between the ground and altitudes of up to hundreds of meters above ground, in which other mobile platforms have been unable to operate (Shaw et al., 2021). Due to their relatively low flying speeds, UAV platforms offer a high spatiotemporal resolution for sampling and thus enabling accurate plume mapping. On the other hand, UAVs have limited endurance, being constrained by battery capacities and payloads, making them more suitable for small facility flux quantification.

UAV platforms have been used to quantify $CH_4$ emissions in several studies, mainly focused on facility-scale
emission sources including landfills (Allen et al., 2019; Bel Hadj Ali et al., 2020), coal mines (Andersen et al., 2021),
dairy farms (Vinkovic et al., 2022), wastewater treatment plants (Gålfalk et al., 2021) and oil and gas facilities (Golston
et al., 2018; Li et al., 2020; Nathan et al., 2015; Shah et al., 2020; Tuzson et al., 2021). UAV-based $CH_4$ measurements
are generally made with three different methods: collecting on-board samples for subsequent analysis, tethered sampling
to a sensor on the ground and on-line measurements (Shaw et al., 2021). Gas samples could be stored onboard a UAV for
subsequent analyses on the ground after landing, using air bags (Brownlow et al., 2016) or sampling canisters (Chang et
al., 2016). Andersen et al. developed a UAV-based active AirCoresystem, consisting of a long coiled stainless-steel tubing,
a small pinhole orifice, and a pump that drags air through the tube (Andersen et al., 2018), which allows for a higher
spatiotemporal resolution in the measurements. Direct comparisons between a quantum cascade laser absorption
spectrometer (QCLAS) and the active AirCore measurements show that the active AirCore measurements are smoothed
by 20 s and had an average time lag of 7 s. The active AirCore measurements also stretch linearly with time at an average
rate of 0.06 s for every second of QCLAS measurement (Morales et al., 2022). The advances in active AirCore sampling
have made UAV measurements for $CH_4$ emissions feasible, even if still with rooms for improvement. Studies of using
UAVs for $CO_2$ plume detection and mappingfrom anthropogenic sources have also been reported (Reuter el al., 2021;
Liu et al., 2022; Leitner et al., 2023; Chiba et al., 2019). Reuter et al. presented the development of a UAV platform  to
quantify the $CO_2$ emissions of anthropogenic point sourcrs by deployment of an NDIR (non-dispersive infrared) detector
and a 2-D ultrasonic acoustic resonance anemometer on the platform (Reuter et al., 2021).
In this study, we developed a new active air sampling system for deployment from a UAV  on a trajectory in the
three-dimensional space to measure $CO_2$ and $CH_4$. The complete sampler plus UAV system was deployed to quantify
$CO_2$ and $CH_4$ emissions from the stacks of the main coking plant of Shagang, the largest private steel maker in China.
The top-down emission rate retrieval algorithm (TERRA) (Gordon et al., 2015) was applied to the UAV data to determine
stack $CH_4$ and $CO_2$ emissions rates. The iron and steel industry is one of the largest contributing industries to global GHG
emissions, accounting for around 7% of global total GHG emissions (Hasanbeigi, 2022). Coke production is one major
process of iron and steel making that generate emissions of $CO_2$ and $CH_4$. During coke production, coking coal is used to
manufacture metallurgical coke that is subsequently used as the reducing agent in the production of iron and steel (U.S.
Environmental Protection Agency, 2016). Coke oven gas is the main sources of $CO_2$ and $CH_4$ emissions during coke
production (Angeli et al., 2021; IPCC, 2006). China is the largest coke producer in the world, with a coke production of
4.72 billion tons in 2020. The GHG emissions from coke production in China are reported based on the Tier 1
methodology of IPCC Guidelines, which multiplies generic default emission factors with the tonnage of coke produced
(Ministry of Ecology and Environment of China, 2018). Tier 1 methodologies are the simplest and least complex requiring
less resources on collecting the necessary data and producing GHG emission estimates. The present UAV measurement-
based emission results can be compared with material balance based emission estimates and the emissions based on the
Tier 1 emission factors and coke production at the plant, and to shed light on the uncertainties related to Tier 1 emission
factors in the case of $CH_4$ emissions.
**2.Method**
**2.1 The air sampling system**
To realize GHG emission quantification by UAV measurement, a new compact air sampling system was developed based
on a variation of the active AirCore method. The AirCore system contains a 150-m-long stainless steel tube, open at one
end and closed at the other, that relies on positive changes in ambient pressure for passive sampling of the atmosphere
(Karion et al., 2010). Figure 1 shows an overview of the patented design for this sampler. It consists of a 150 m long thin-
walled 1/8 inch outside diameter stainless-steel tubing, a pump, a micro-orifice, a $CO_2$ marker generator, two three-way
solenoid valves and electric relays, with all electrical devices powered by a 12V battery. The tubing is winded into a
multilayer coil, in whose center the other components of the system are mounted. The system is housed in the highly
compact patented carbon fiber assembly design of 280 mm diameter and 98 mm height, that can be quickly mounted at
and dismounted from the bottom of an UAV. The sampler weighs about 5.9 kg and allows for continuous sampling up to
35 minutes.

101         The sampler air intake is mounted at 70 cm above the center of gravity of the UAV, placed nearby a sonic

anemometer (below) for ensuring sampling the same air mass where wind speed is measured. The time stamp of the
mixing ratio observation was corrected for the short time lag of 4 seconds between sampling at the air intake and the thin-
walled stainless-steel tubing attributable to the length of the Teflon inlet tube. Shortly before every flight, the pump is
remotely turned on to sample the $CO_2$ marker for 5 seconds and then to collect air samples. The $CO_2$ markers help to
identify the starting point and specific times subsequently during the UAV air sampling in data extraction and analysis.
During flight, the pump would alternatively sample the marker and the ambient air on a preset timing schedule. The
sampling flow rate remains at 18 ccm during the entire flight, controlled with the micro-orifice which is placed between
the pump and the coiled tubing. After landing, the pump is remotely turned off and the air sample in the sampling tubing
is immediately analyzed with a cavity ring-down spectrometer (CRDS) (Picarro, Inc., CA, USA, model G2401) for $CO_2$
and $CH_4$ mixing ratios in the sampled air. Waiting longer would lead to unwanted mixing of the samples in the tubing.
The air sample enter the tubing from the air inlet during sampling and leave the tubing from a different air outlet during
later analysis. As a result, the samples at the beginning of the flight spend the same amount of time within the tubing as
those at the end of the flight.  Using the embedded $CO_2$ marker data, the $CO_2$ and $CH_4$ data series can be mapped to the
sampling times and GPS locations during flight.

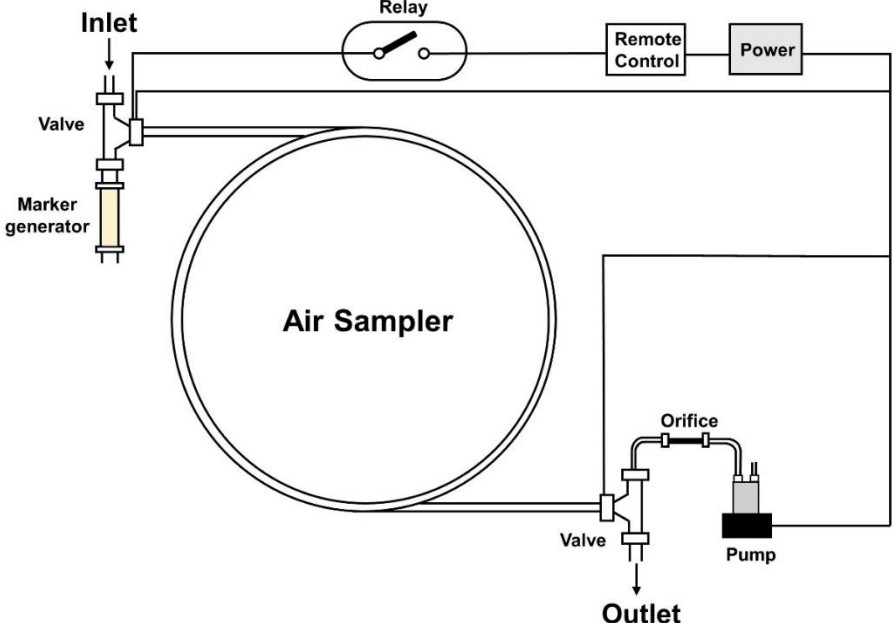

**Figure 1.** Design of the air sampler.
**2.2 The 3D sonic anemometer**
Previous studies that applied UAV platforms for GHG monitoring generally relied wind data from nearby ground weather
stations (Morales et al., 2022; Allen et al., 2019). However, Gålfalk et al. shows that wind speeds were inconsistent
between a ground weather station at a 1.5 m height and an anemometer mounted on their UAV, especially when altitude
increases, showing the need to have an on-board weather station for accurate flux calculations (Gålfalk et al., 2021). In
the present study, in order to obtain meteorological data along the flight track, a 3D sonic anemometer (Geotech Inc,
Denver, US, model Trisonica Mini) is attached on the top of the UAV via a 450 mm carbon fiber pole. The anemometer
measures three-component wind speed ($U_x$, $U_y$, $w$) and temperature ($T$). The measured data were further transformed into
actual wind speeds and wind directions after corrections for UAV attitude (pitch, yaw, roll) changes and accounting for
its airspeed, as well as the perturbations caused by the UAV rotor propellers using a patented correction algorithm. The
GPS information, airspeed, and attitude data (pitch, yaw, and roll) were extracted from the UAV data transmitted to the
ground control station. The anemometer measures wind speeds within the range of 0 to 50 m s$^{-1}$, with an accuracy of $\pm$
0.1 m s$^{-1}$ below the wind speed of 10 m s$^{-1}$. The accuracy for wind direction measurement is $\pm$ 1°. For temperature
measurement, the operating range for the anemometer is between -40 °C to 85 °C and the accuracy is $\pm$ 2 °C.
For anemometers mounted on multi-rotor UAVs, how to correct for the effects of the translational and rotational
movements of the UAVs as well as the flows induced by the rotors to obtain accurate wind data is an on-going research
topic (Gålfalk et al., 2021; Wolf et al., 2017; De Boisblanc et al., 2014; Palomaki et al., 2017; Zhou et al., 2018). During
flight, rotary wing UAVs create thrust by drawing air from above the rotors and expelling it downwards at a higher
velocity. Such flows may extend to the anemometer position in addition to true atmospheric air flows, masking the true
wind signals in the data from the anemometer (Wolf et al., 2017). Previous studies have conducted laboratory testing
(Wolf et al., 2017; De Boisblanc et al., 2014; Palomaki et al., 2017) or flow field simulation (Zhou et al., 2018) to
determine the appropriate distance to place anemometers onto multi-rotor UAVs to minimize the impact from the rotor-
induced air flows. The anemometer in this research is mounted at an upward distance of 70 cm from the center of gravity
of the UAV. A full digital model of the UAV, the anemometer and its mounting frame, and the air sampler was created.
Using this digital model, computational fluid dynamics (CFD) simulations were performed to quantify wind speed
disturbances caused by the UAV's rotor propellers on the anemometer during flight under a vast array of different wind
conditions. An overall correction algorithm was developed in which parameters for propeller disturbances determined
based on the CFD simulations were included along with correction schemes for false signals resulting from translational
motions and changes in UAV pitch, roll and yaw.
**2.3 The UAV**
The air sampler and the anemometer are mounted on a hexacopter UAV (KWT-X6L-15). The UAV has a maximum flight
time of ~30 minutes at a maximum payload of 15 kg, or longer with a lighter payload. Such flight endurance and carrying
capacity meet our needs for loading the air sampler and the anemometer onto the UAV to realize emission quantification.
The UAV is capable of flying at winds up to 14.4 m s$^{-1}$ to an altitude of about 4000 m and has a maximum horizontal
flying speed of 18 m s$^{-1}$, a maximum ascending speed of 4 m s$^{-1}$ and a maximum descending speed of 3 m s$^{-1}$. The
horizontal hovering precision of the GPS on the UAV is $\pm$ 2 m and the vertical hovering precision is $\pm$ 1.5 m.

**2.4 Air sample analysis**

After landing, the air sample collected in the tubing is immediately analyzed with the CRDS analyzer. The withdrawal

flow rate of the air from the sample tubing during analysis is an important parameter in optimizing the results. High

withdrawal rates lead to unwanted mixing in the cavity of the analyzer. However, direct withdrawal of air from the sample

tubing by the analyzer at a flow rate as low as the sampling flow rate of 18 sccm results in smoothing of concentrations

from the inner wall surface drag and desorption inside the tubing. We optimized the flow rate of the air from the sample

tubing into the CRDS analyzer at $\sim$ 54 sccm, 3 times the sampling flow rate, by diluting the air sample with zero air, with

two mass flow controllers separately controlling the flow rate of zero air and the withdrawal rate of the air sample (Fig.

2b).

**2.5 Mass balance approaches for determining emission rates**

The UAV-based measurements were coupled with the mass-balance approach TERRA to determine the emission rates of

the measured pollutants using their measured mixing ratios and the meteorological data (three-component wind speed

$(U_x, U_y, w)$ and temperature $(T)$) collected on board the UAV during the flight. TERRA computes integrated mass fluxes

through airborne virtual box/screen measurements including those made from aircraft and in this case UAVs. TERRA

has been used successfully and extensively for emission rate determination of tens of volatile organic compounds (Li et

al., 2017), $CO_2$ (Liggio et al., 2019), $CH_4$ (Baray et al., 2018), oxidized sulphur and nitrogen (Hayden et al., 2021), black

carbon (Cheng et al., 2020) and secondary organic aerosol (Liggio et al., 2016) using aircraft measurements. To run

TERRA based on a virtual box flight, the first step is to map the $CH_4$ and $CO_2$ mixing ratio data measured along the level

flight tracks encircling a facility to the two-dimensional virtual walls of the virtual box, created from stacking the level

flight tracks, that surrounds the facility. The two-dimensional virtual walls (or screens) are derived from the unwrapping

of the virtual box, to assist the presentation of the $CH_4$ and $CO_2$ plumes along the flight tracks, with the horizontal path

length (i.e., the ground line projection of the fitted flight track) and altitude as the two dimensions. The start of the

horizontal path is typically defined as the south-east corner of the virtual box, but the selection of this starting position

has no effect on the emission rate computation, and the horizontal path distance increases in a counter clockwise direction.

This procedure results in a translation of each flight position point from a three-dimensional positionof latitude ($y$),

longitude ($x$), and altitude ($z$, above mean sea-level) to a two-dimensional screen position of horizontal path distance $s =$

$f(x,y)$. Subsequently, TERRA applies the Simple Kriging algorithm to interpolate the data and produces a mesh on the

two-dimensional virtual box walls whose resolution can be set depending on applications. The kriging weights were

obtained with an isotropic spherical semivariogram model. In TERRA, nugget, sill, and range can all be modified to fit
the semivariogram model. The mixing ratios of both $CH_4$ and $CO_2$ are extrapolated from the lowest flight altitudes to the
ground digital elevation using one of several methods or a combination thereof, namely (1) assuming a constant (2) linear
extrapolation between a constant and background (3) a background value below flight altitudes (4) linear fit between the
lowest flight altitude and zero at the ground and (5) exponential fit from the lower flight altitudes (Gordon et al., 2015).
Concurrently measured wind speed from the UAV is decomposed into northely and easterly components
$(U_N(s,z), U_E(s,z))$ based on the wind direction and similarly interpolated onto the 1 m x 2 m mesh. The decomposed
wind speeds are further extrapolated to the ground digital elevation using a log profile fit (Gordon et al., 2015). Based on
the interpolated/extrapolated $CH_4$ and $CO_2$ mixing ratio, temperature, pressure (calculated using barometric height
formula), and wind speeds, TERRA computes the fluxes of $CH_4$ and $CO_2$ through the virtual walls and finally their facility
emission rates by integrating the fluxes.
To summarize, in TERRA the mass-balance in computing the emissions within a control box for a given inert
pollutant such as $CH_4$ or $CO_2$ is presented by:
$$E_C = E_{C,H} + E_{C,V} - E_{C,M} \; , \tag{1}$$
where $E_C$ is the emission rate, $E_{C,H}$ is the horizontal advective transfer rate through the box walls, $E_{C,V}$ is the advective
transfer rate through the box top and $E_{C,M}$ is the increase in mass within the volume due to a change in air density. Other
terms listed in the Gordon et al. computation algorithm that were used to solve for the total emission rate were often
neglected as they contribute little to the total emission rates (Gordon et al., 2015). Each term from Eq. (1) is estimated as:
$$E_{C,H} = M_R \iint X_C \rho_{air} U_\perp \mathrm{d}s\mathrm{d}z \; , \tag{2}$$
$$E_{C,V} = M_R X_{C,Top} \iint \rho_{air} \omega \mathrm{d}x\mathrm{d}z \; , \tag{3}$$
$$E_{C,M} = M_R \iiint X_C \frac{\mathrm{d}\rho_{air}}{\mathrm{d}t} \mathrm{d}x\mathrm{d}y\mathrm{d}z \; , \tag{4}$$
where $M_R$ is the ratio of the compound molar mass to the molar mass of air, $X_C(s,z)$ is the mixing ratio of the compound
in question, $\rho_{air}(s,z)$ is the air density, $w$ is the vertical wind velocity at the box top, $X_{C,Top}$ is the mixing ratio at the top
of the box, and $U_\perp(s,z)$ is the horizontal normal wind vector to the flight track calculated from the northely and easterly
components $(U_E(s,z), U_N(s,z))$:
$$U_\perp(s,z) = \frac{U_N(s,z)\mathrm{d}s/\mathrm{d}x - U_E(s,z)\mathrm{d}s/\mathrm{d}y}{\sqrt{(\mathrm{d}s/\mathrm{d}x)^2 + (\mathrm{d}s/\mathrm{d}y)^2}} \,,$$ (5)
The vertical transfer rate term $E_{C,V}$ is estimated by computing the air mass vertical transfer rate, determined from air
mass balance within the box, and multiplying it with the $CO_2$ or $CH_4$ mixing ratios at the box top. This term is normally
negligible in other top-down emission estimate approaches since it is typically miniscule compared to horizontal fluxes,
but can affect the computed emission rates when vertical air movement becomes more significant such as under unstable
atmospheric conditions. $E_{C,M}$ is often ignored in other mass-balance approaches; in TERRA it is estimated by taking the
time derivative of the ideal gas law in temperature and pressure during the flight time, and typically it does not change
significantly over the duration of 30 minutes or so for the UAV flight.
To suit the UAV measurements, the following modifications to the TERRA algorithm were made: (1) A much higher
interpolation resolution for the kriging mesh was implemented for application to the UAV measurements in this study,
with the interpolation mesh size adjusted to 1 m (vertical) by 2 m (horizontal), as UAVs fly significantly shorter distances
compared to applications to piloted aircraft for which the interpolation resolution was 20 m (vertical) by 40 m (horizontal);
(2) The modified TERRA now applies an embedded routine to automatically fit flight tracks using least squares, while
this procedure was previously conducted manually offline through geographic information system when using TERRA.
(3) The modified version of TERRA has added an algorithm for correcting negative weights during Kriging interpolation
following Deutsch (Deutsch, 1995). TERRA has been updated at Peking University now recoded using the Python
language and runs under a browser-server environment with a new GUI and new interactive data flow.
**3 Laboratory tests**
**3.1 Validation of the air sampler**
Prior to flights in the field, we validated the air sampler in laboratory experiments by first sampling an artificial air while
making simultaneous online measurements of the artificial air with the CRDS analyzer, and then analyzing the sampled
artificial air was with the same CRDS analyzer and comparing the results from the air sampler to the online measurements.
An experimental apparatus was constructed for the simultaneous sampling of the same artificial air with the air sampler
and the CRDS analyzer through a tee junction (Fig 2(a)), and subsequent air sample analysis using the same CRDS
analyzer (Fig. 2(b)). In the artificial air, $CH_4$ and $CO_2$ standards were control-released into the lab air from an 8 L gas
cylinder filled with a gas mixture of 5 ppm $CH_4$, 2 ppm CO and 600 ppm $CO_2$ to generate the artificial air source. The
outlet of the standard gas cylinder was held at varying distances to the tee junction over time to yield a time series of
different $CH_4$ and $CO_2$ mixing ratios, which was designed to mimic plumes expected in the real atmosphere. During
analysis, the flow rate through the zero air (Mass Flow Controller 1) is adjusted to make sure that the flow rate through
the air sampler (Mass Flow Controller 2) is stable and consistent at 54 sccm (Sec. 2.4).

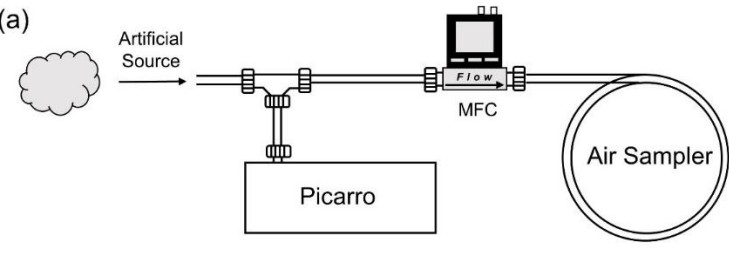

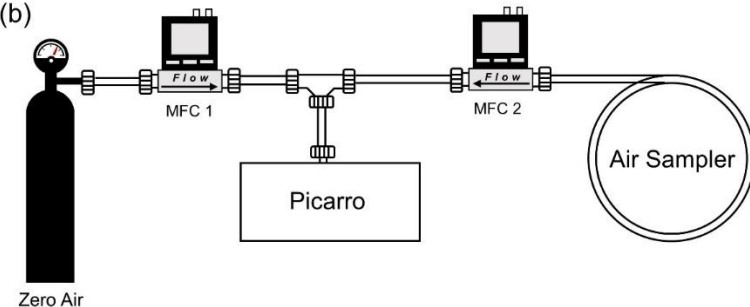


**Figure 2.** Diagram of the air sampler testing setup in the laboratory. (a) simultaneous sampling by the air sampler and the Picarro
CRDS analyzer. (b) subsequent air sample analysis using the picarro CRDS analyzer.
Figure 4(a) illustrates the mixing ratios of $CO_2$ and $CH_4$ time series obtained from the air sampler and online
measurements by the CRDS analyzer. It can be seen that the measured results from the air sampler and the online CRDS
measurements analyzer are in good agreement throughout the tests, and the correlation coefficient is estimated to be 0.89
and 0.73 for $CH_4$ and $CO_2$ (Fig. 4(c) and (f)). For the measurements with the air sampler, short term variations and noises
in the $CH_4$ and $CO_2$ mixing ratios, that were fully captured by the CRDS analyzer during the online measurements, were
smoothed out, while the main features and tendencies were preserved. In fact, the air sampler measurement results should
be a smoothed version of the CRDS analyzer online measurements, due to mixing in the analyzer cavity, molecular
diffusion during sample storage in the sampler, inner wall surface drag and desorption during its withdrawal from the
tubing during analysis, as well as Taylor dispersion during sampling and analysis (Karion et al., 2010). Dilution with zero
air during later CRDS analysis also contributes to the smoothing.

**3.2 Data deconvolution to achieve high time resolution**

While it is impractical to delineate the individual smoothing effects when the air sample passes through the coupled system of the sampler plus the analysis setup as described above, the measured concentration $y(t)$ can be treated as a result of the convolution of the air concentration before sampling $x(t)$ and a smoothing kernel $g(i)$ consisting of a series of weights, which are inherently determined by factors including the sampler properties (tubing length, inner diameter, temperature, absorptive properties, flow rates), storage time, dilution, and mixing in the cavity of the instrument. The smoothing can be described as:

$$y(t) = \sum_{i=r}^{s} g(i)x(t-i) + n(t), t = s, s+1, \ldots, n-1+r, \tag{6}$$

Or, expressed as a convolution of the form:

$$y(t) = g(t) * x(t) + n(t), \tag{7a}$$

where $y(t)$ is the measured concentration at time t, $x(t)$ the air concentration, and $n(t)$ the unknown noise, assumed to be independent of $x(t)$. The kernel $g(i)$ contains $s - r + 1$ non-zero kernel weight terms $(0 < g(i) < 1)$. When all four terms in Eq. (7a) undergo Fourier transform, Eq. (7a) can be expressed in the frequency domain:

$$Y(f) = G(f)X(f) + N(f), \tag{7b}$$

In order to characterize the kernel weights $g(i)$, a second lab experiment was conducted during which the sampler first sampled zero air for some time, and then sampled the $CO_2$ and $CH_4$ standards for one second, before returning to sampling zero air again, creating an original concentration pulse signal in the $x(t)$:

$$x(t) = \begin{cases} C, & t = j \\ 0, & t \neq j \end{cases}, \tag{8}$$

where $j = j^{th}$ second when the sampler collected the standard of a known concentration C. This air sample was then analyzed with the CRDS as described above. After sampling, storing and analyzing, smoothing of the original concentration pulse leads to the concentration signal output $Y(t)$ as follows:

$$y(t) = \begin{cases} \sum_{i=r}^{s} g(i)x(t-i) + n(t) = g(t-j)C + n(t), & t-i = j \text{ and } i = r, r+1, \cdots, s \\ n(t), & t-i \neq j \end{cases}, \tag{9}$$

where $y(t)$ is the measured concentrations from the air sampler after sampling the concentration pulse and is non-zero when $t - i = j$, with the index $i$ taking the values from $r$ to $s$. The noise $n(t)$ term is zero for $t - i \neq j$ and can be assumed to have similar behavior for $t - i = j$. Therefore,

$$g(i) = g(t-j) = \frac{1}{C}y(t) - \frac{1}{C}n(t), \ t = i + j \text{ and } i = r, r+1, \cdots, s, \tag{10}$$

The second lab experiment showed that $y(t)$, and therefore the kernel $g(t)$, consists of 70 non-zero values. To remove
the noise $n(t)$, $g(t)$ is further smoothed using a box-car running mean of 5 terms:
$\hat{g}(t) = \frac{1}{5}\sum_{k=t-2}^{k=t+2} g(k) \approx \frac{1}{C} y(t),\ t = i + j\ and\ i = r, r + 1, \cdots, s$,  (11)
It could be seen from Fig. 3 that $\hat{g}(t)$ has an asymmetrical distribution with a right trailing tail and a half-height width of
approximately 20 seconds for $CO_2$ and 21 seconds for $CH_4$, indicating that the smoothing had significantly reduced the
sampling/analysis method time resolution to about 20 second from the 1 second resolution of the original pulse in the air
concentration. The kernel shows that the influence from the neighboring points have on a given point decreases with
increases in the gap between the two points.

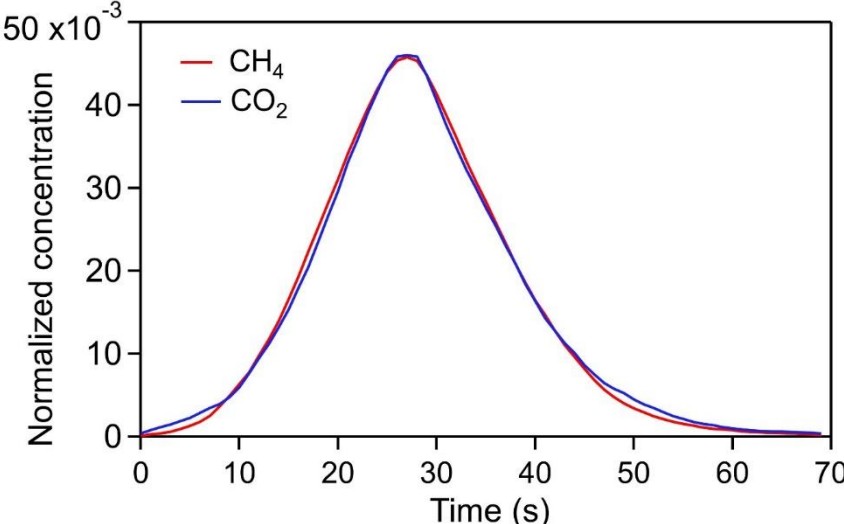


**Figure 3.** The output of the one-second signal after sampling, storing and analyzing using the air sampler for $CO_2$ and $CH_4$, normalized
by their respective concentrations in the standard. As shown in the text, these curves are the actual kernel weights of $\hat{g}(t)$.
To test whether the kernel weights $\hat{g}(t)$ can smooth the online measured concentrations from the first lab experiment
(top line in Fig. 4(a) and (b)), the weights $\hat{g}(t)$ were used to convolute with the data from the online measurements (i.e.,
$x(t)$), resulting in an estimated $\hat{y}(t)$ (Fig. 4(a) and (b), third line) that is in excellent agreement with the measurements
from the air sampler, with the correlation coefficients increased to 0.99 and 0.98 for $CH_4$ and $CO_2$ (Fig. 4 (d) and (g)).

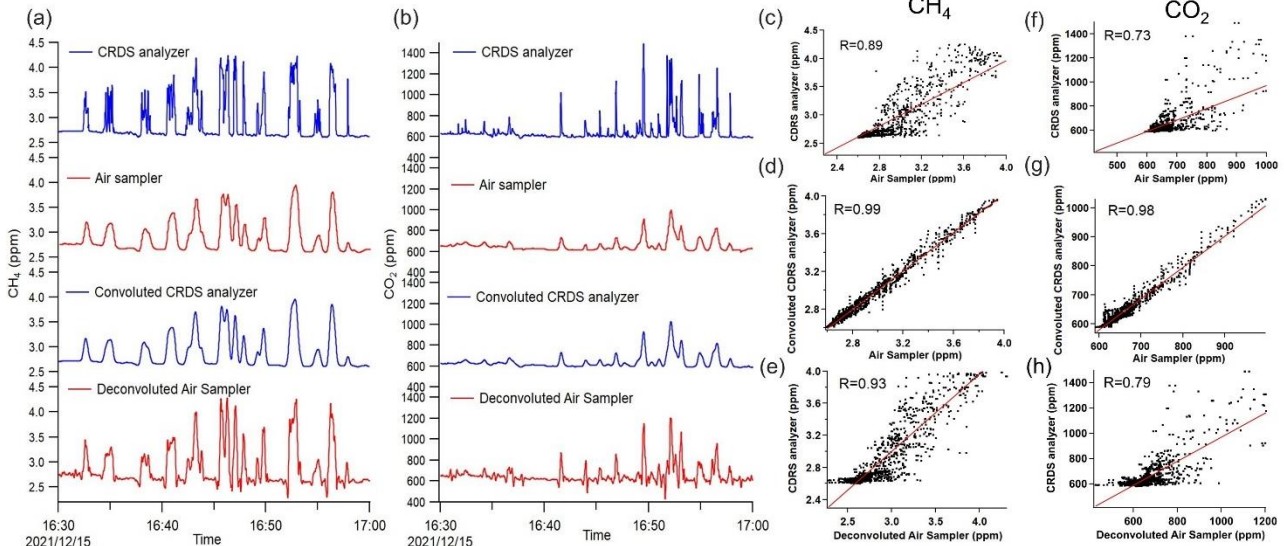


**Figure 4.** (a) and (b) Mixing ratios of $CO_2$ and $CH_4$ measurements by online measurements with CRDS (the first line) and the air sampler(the second line) in laboratory tests. The third line represents the smoothed CRDS data after convolution with the kernel $\hat{g}(t)$ and the fourth line represents the deconvoluted series after Wiener deconvolution. The signals of the same color represent the original signals and the corresponding signals after convolution or deconvolution (c)-(e) Correlation plots of $CH_4$ (f)-(h) Correlation plots of $CO_2$.

The ultimate goal of determining $\hat{g}(t)$ in Fig. 3 is to deconvolve $y(t)$ from the air samplerto obtain the original concentration series $x(t)$ using a number of deconvolution techniques. In the present study, we used the deconvolution method based on the Wiener theorem (Lin and Jin, 2013). The theorem provides the Wiener convolution filter $h(t)$ so that $x(t)$ can be estimated as follows:

$$\hat{x}(t) = \sum_{i=-\infty}^{\infty} h(i)y(t-i) = h(t) * y(t), \tag{12}$$

where $y(t)$ is the measured concentration, and $\hat{x}(t)$ an estimate of $x(t)$. In the frequency domain, Eq. (12) may be rewritten as a product of two scalars:

$$\hat{X}(f) = H(f)Y(f), \tag{13}$$

where $\hat{X}(f)$, $H(f)$, and $Y(f)$ are the Fourier transforms of $\hat{x}(t)$, $h(t)$, and $y(t)$, respectively. The Wiener convolution filter $h(t)$ is derived from the minimization of the mean square error:

$$\epsilon(f) = E\left|X(f) - \hat{X}(f)\right|^2, \tag{14}$$

with $E$ denoting the expectation. When Eq. (7b) and Eq. (13) are substituted into Eq. (14) and the quadratic is expanded, the mean square error $\epsilon(f)$ can be differentiated with respect to $H(f)$ and the derivative $\frac{d\epsilon(f)}{dH(f)}$ is set to zero to achieve the minimization; under the assumption that the noise $N(f)$ is independent of $X(f)$, $H(f)$ is derived as

$H(f) = \frac{G(f)S(f)}{|G(f)|^2 S(f) + N(f)}$ , (15)
where $G(f)$ is the Fourier transform of $\hat{g}(t)$ derived from the second lab experiment described above, $S(f) = E|X(f)|^2$
and $N(f) = E|N(f)|^2$ are the mean power spectral densities of the original concentration series $x(t)$ and the noise $n(t)$,
respectively. Equation (15) could be rewritten as:
$H(f) = \frac{1}{G(f)} \left[ \frac{|G(f)|^2}{|G(f)|^2 + N(f)/S(f)} \right] = \frac{1}{G(f)} \left[ \frac{|G(f)|^2}{|G(f)|^2 + 1/SNR(f)} \right]$ , (16)
where $SNR(f) = S(f)/N(f)$ is the signal-to-noise ratio.
Substituting Eq. (16) into Eq. (13), $\hat{X}(f)$, the Fourier transforms of $\hat{x}(t)$, is derived. The deconvolution is completed
with the inverse Fourier transform of $\hat{X}(f)$ to give $\hat{x}(t)$, the estimated air concentrations. The deconvolved series of $CH_4$
and $CO_2$ restored with the Wiener convolution filter are shown in Fig. 4(a) and (b), and the correlation coefficient between
the deconvoluted results and the online measurements with the CRDS analyzer are 0.93 and 0.79 for $CH_4$ and $CO_2$ (Fig.
4 (e) and (h)), higher than that between the original air sampler measurement and the CRDS analyzer. These results
indicates the effectiveness of the Wiener theorem to deconvolve a smoothed series to a much higher time resolution while
accounting for noise. The restored series is improved in terms of time resolution, from about 20 seconds mentioned above
to about 3~4 seconds after the deconvolution. The lab test data from the online measurements contain strong high-
frequency components, artificially manipulated to provide an extreme case for testing the deconvolution algorithm. Such
high frequencies lead to some residual noise in the deconvolved results, primarily as a result of choosing the cutoff
frequencies for the mean power spectral densities $S(f)$ and $N(f)$. Nevertheless, such a situation will be improved for
sampling in the real atmosphere where sub-second high-frequency variations are not common.
**4. Field application**
To apply the UAV-based measurement system described above to atmospheric measurements of $CO_2$ and $CH_4$,
flights were made at the Shagang Group located in Jiangsu, China on 28 December 2021. Shagang Group is a major iron
and steel company on the south shore of the Yangtze River (31.9704° N, 120.6443° E). The company produces over 40
million tons of steel each year, making it one of China's top-five steel producers. Onsite coke making for iron production
is located in the western part of the Shagang Steel complex. The coke making process is to dry distill coal in a coking
oven at ~1000 °C temperature to boil off volatile components to form coke (metallic coal). During coke production,
combustion of coking oven gas, blast furnace gas from steel making, and coal tar plus light oil for heating the coking
oven is the main $CO_2$ and $CH_4$ emission source.
Two coking plant stacks were chosen as the target emission source for the field UAV flight. During flight, the UAV
was flown in a rectangle pattern (200 m×500 m) that encloses the two stacks, with repeated flight tracks at 9 altitude
levels that, when stacked, created a virtual box and intercepted the emitted $CO_2$ and $CH_4$ plumes on the downwind side
of the box. The UAV ascended from the ground to 135 m a.g.l. and started the box flight at this altitude, ascending 15 m
every level and reaching a maximum altitude of 255 m a.g.l. before landing. The UAV maintained a constant horizontal
speed of 8 m s$^{-1}$ during flight. The flight lastd for approximately 30 minutes. It's assumed that the plume remains steady
during the time of measurement. After landing, the air sample collected in the sampler was immediately analyzed with
the CRDS analyzer as per the procedure described above in Fig. 2(b).
**5. Result and discussion**
**5.1 $CH_4$ and CO2 mixing ratio enhancement from the coking plant**
Figure 5(a) shows the time series of $CH_4$ and $CO_2$ mixing ratios measured with the air sampler at the coking plant during
the flight (red line). The air sampler sampled for a total of 30 minutes during the flight. After landing, the air sample was
analyzed for 10 minutes, as the analysis flow rate triples the sampling flow rate (54.0 sccm vs. 18.0 sccm). The time scales
of instrument readings were then stretched three times to restore the original time scales. The $CH_4$ and $CO_2$ time series
were then deconvolved using the convolution kernel obtained from laboratory test (Sec. 3.2) to restore the mixing ratio
time series in air (black line). The meteorological parameters during the time of flight were measured by the 3D
anemometer, showing consistent southwesterly winds (Fig. 5(b)). The average windspeed is 4.7±4.9 m s$^{-1}$ and the average
winddirection is 216.4±38.4° during the time of flight.  Consistency of wind measurements can be seen from the two
wind rose plots for the northern wall and the southern wall respectively. During the flight, the maximum mixing ratio
measured was 5.6 ppm for $CH_4$ and 1356 ppm for $CO_2$. During the 30-minute flight, a total of 5 $CO_2$ makers were
generated during the 30 minutes of sampling (Fig. 5(a)), and the decreases in the marker concentrations are corrected with
a Gaussian form function.

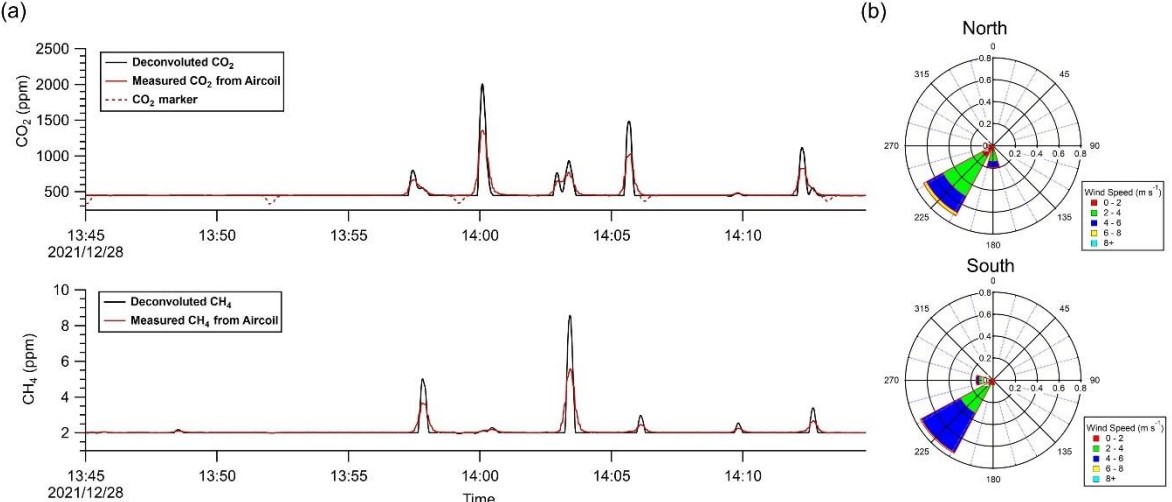


**Figure 5.** (a) Red line represents $CH_4$ and $CO_2$ mixing ratios measured from the air samples collected with the air sampler during the
flight at the coking plant. Black line represents the deconvolved $CH_4$ and $CO_2$ time series and red dashed line sections represent the
original marker $CO_2$ concentrations every 7 minutes. (b)Wind rose plot for the northern and southern wall based on the onboard
meteorological measurements during the flight.
**5.2 Emission estimation**
The $CO_2$ and $CH_4$ emission rates for the stacks from coking plant were estimated by applying a version of the computation
algorithm TERRA specifically modified to suit UAV measurements. The deconvolved mixing ratio time series of $CO_2$
and $CH_4$ were used in the TERRA algorithm. The algorithm first maps the mixing ratios to the walls of the virtual box,
then applies a kriging scheme to interpolate the data and produces a 2 m (vertical) by 1 m (horizontal) mesh on the virtual
box walls (200m×500m) (Fig. 6). The semivariogram of the flight points was fitted with a spherical model (range=300,
sill=3, nugget=0). Wind speed and wind direction are first decomposed into northly and easterly components, then further
converted to vectors that are normal to and parallel to the walls of the virtual box before kriging. Background $CH_4$ and
$CO_2$ were determined using upwind measurements. The background between upwind data was linearly interpolated and
box-car smoothed within a 3-4 minute moving window to derive a variable baseline $CH_4$ and $CO_2$ for the entire 30-minute
flight. As shown in Fig. 6, the $CH_4$ and $CO_2$ plumes can be seen at different locations on the downwind side of the box
wall, which indicates that the $CH_4$ plume and the $CO_2$ plume probably came from different sources within the box. Using
the modified version of TERRA, the emission rates for the two stacks in the coking plant were calculated to be 0.12 ±
0.01 t h$^{-1}$ for $CH_4$ and 110 ± 20 t h$^{-1}$ for $CO_2$. The uncertainties for the estimates were derived from detailed analyses of
each uncertainty source including measurement error in mixing ratio and wind speed, the near-surface wind extrapolation,
the near-surface mixing ratio extrapolation, box-top mixing ratio, box-top height and deconvolution.

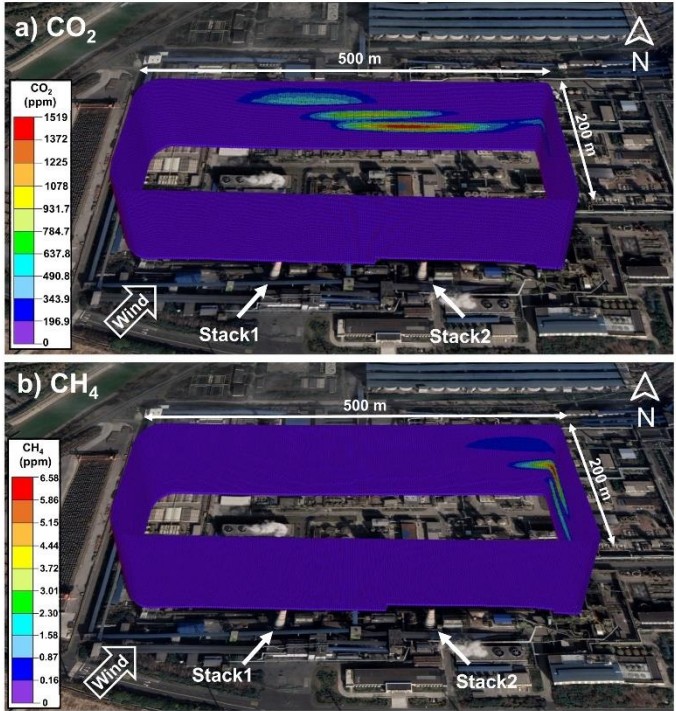


**Figure 6.** Virtual flight box for monitoring $CO_2$ (a) and $CH_4$ (b) during the flight. The $CO_2$ and $CH_4$ plumes were captured on the north
and east wall respectively. The wind came from the southwestern direction. Satellite imagery © Google Earth 2019.

**5.3 Uncertainty Analysis**

To determine the overall uncertainty in the emission rates, each source of uncertainty contributing to the overall
uncertainty needs to be identified and quantified. For the emission rate quantification from UAV measurement, the
sources of uncertainties include: measurement uncertainties in the mixing ratios and wind speeds ($\delta_M$), the near-surface
wind extrapolation ($\delta_{Wind}$), the near-surface mixing ratio extrapolation ($\delta_{Ex}$), box-top mixing ratio ($\delta_{Top}$), box-top height
($\delta_{BH}$) and uncertainties due to data deconvolution as shown in the main text ($\delta_{Deconv}$). Each uncertainty is treated as an
independent estimate, and all uncertainties are propagated in quadrature to determine the overall uncertainty in the
estimated emission rate:

$$\delta^2 = \delta_M^2 + \delta_{Wind}^2 + \delta_{Ex}^2 + \delta_{Top}^2 + \delta_{BH}^2 + \delta_{Deconv}^2 , \qquad (17)$$

The accuracy of the mixing ratio measurements from the Picarro CRDS analyzer is 50 ppb and 1 ppb for $CO_2$ and
$CH_4$, respectively. By adding variations in the measured mixing ratios based on the measurement accuracies and re-
applying TERRA, the derived emission rates varied within 1% for both $CO_2$ and $CH_4$. Thus, the uncertainties in the
emission rates due to mixing ratio measurements ($\delta_M$) were estimated at 1% for both $CH_4$ and $CO_2$.

The anemometer measures wind speeds with an accuracy of $\pm 0.1$ m s$^{-1}$ at wind speeds $< 10$ m s$^{-1}$ and wind directions
with an accuracy of $\pm 1°$. The uncertainty of the wind measurements ($\delta_{Wind}$) was estimated using error propagation in the

normal wind $U_\perp(s,z)$, as it is calculated from the northerly and easterly wind components, thus from wind speed and
wind direction:
$$\delta_{U_\perp} = \sqrt{\delta_{\text{easterly}}^2 + \delta_{\text{northly}}^2 + 2\sigma_{\text{easterly}-\text{northly}}} \, , \tag{18}$$
$$\delta_{\text{easterly}} = |\text{WS}\cos(\text{WD})\sigma_{\text{WD}}| \, , \tag{19}$$
$$\delta_{\text{northly}} = |\text{WS}\sin(\text{WD})\sigma_{\text{WD}}| \, , \tag{20}$$
Using this calculation, the uncertainty of the normal wind $\delta_{U_\perp}(s,z)$ was derived at each location. The uncertainty
contributed to the total emission rates to the overall computed emission rate was examined by setting the normal wind to
its upper and lower bounds defined by its uncertainty range, followed by computing the emission rates using TERRA.
The derived $CH_4$ and $CO_2$ emission rates varied by 1.5% and 1.9% respectively. Hence the uncertainties from wind speed
measurements ($\delta_{\text{Wind}}$) were conservatively estimated to be 2% for both $CH_4$ and $CO_2$.
Due to a lack of near-surface measurements along the box walls, extrapolation of $CH_4$ and mixing ratios from the
lowest flight path (~ 150 m above ground level) to the ground level has been shown to be a source of potentially large
uncertainty within TERRA. The magnitude of the uncertainty depends on the nature of the emissions; for example, surface
emissions which may not be fully captured by the flight altitude range have higher uncertainties at ≈20%, whereas elevated
stack emissions which are fully captured by flight altitude range lead to much smaller uncertainties of <4% in the emission
estimates (Gordon et al., 2015). In the present study, to estimate uncertainties due to extrapolating mixing ratios from the
lowest flight track to the ground ($\delta_{\text{Ex}}$), results from all extrapolation techniques (i.e., linear to the ground, constant value
to the ground, linear to background value, or some combination of methods) were derived and compared with the result
using a background value below flight altitudes. Therefore, this term of uncertainty was evaluated at 2% and 6% for $CH_4$
and $CO_2$ respectively.
**Table1.** Emission rates derived using different extrapolation techniques

| Extrapolation techniques | All background below flight altitude (this study) | Constant value from lowest flight altitude to surface | Linear between constant and background at surface | linear | exponential |
|---|---|---|---|---|---|
| $CH_4$ Emissions(kg h$^{-1}$) | 115.7 | 113.9 | 116.9 | 113.9 | 113.6 |
| $CO_2$ Emissions(kg h$^{-1}$) | 110100 | 109970 | 109400 | 109970 | 103960 |

Additional components contributing to uncertainties in the computed emission rates specific to the box approach
include box-top mixing ratio ($\delta_{\text{Top}}$) and box-top height ($\delta_{\text{BH}}$). The TERRA box approach assumes a constant mixing ratio
at the box top ($X_{C,\text{Top}}$) by averaging the measured value at the top level. The term $\delta_{\text{Top}}$ is determined from the 95%
confidence interval ($2\sigma/\sqrt{n}$) of the interpolated measurements. The calculated confidence interval of the mixing ratio at
the box top is $0.01 \pm 0.13$ ppm for $CH_4$ and $70.1 \pm 89.1$ ppm for $CO_2$. A top average mixing ratio of 0.14 ppm for $CH_4$
and 159.2 ppm for $CO_2$ are set as input parameters to derive resulting uncertainties in the emissions rates. Thus, 106.6 kg
$h^{-1}$ for $CH_4$ and 93760 kg $h^{-1}$ for $CO_2$ were derived. Then, this uncertainty term is conservatively taken as 8% and 16%
for $CH_4$ and $CO_2$.
The uncertainty due to the choice of box height, $\delta_{\text{BH}}$, within TERRA is estimated by recomputing the emission rate
with a reduced box height ($z$) of 100 m. The recalculated emission rate after reducing the box height of 100m is 106.4 kg
$h^{-1}$ for $CH_4$ and 113500 kg $h^{-1}$ for $CO_2$, thus $\delta_{\text{BH}}$ is estimated as 8% for $CH_4$ and 3% for $CO_2$.
For cases that use the air sampling system instead of online measuring instruments, as the $CH_4$ and $CO_2$ time series
measured from the air sampler were deconvoluted to restore the unsmoothed time series before being input into the
TERRA algorithm, it is necessary to account for the uncertainty that comes from such deconvolution as outlined in the
main text. Time series before and after deconvolution were applied to the TERRA algorithm to obtain the total emission
rates. The computations show that emission rates before and after deconvolution vary within 1%, which was taken as the
uncertainty $\delta_{\text{deconv}}$. The assessment of uncertainties for the TERRA-computed emission rates from the coking plant are
listed in Table 2.
**Table 2.** Assessment of percent uncertainties for $CH_4$ and $CO_2$ emission rate estimations. The sources of uncertainties
include: measurement uncertainties in the mixing ratios and wind speeds ($\delta_M$), the near-surface wind extrapolation
($\delta_{Wind}$), the near-surface mixing ratio extrapolation ($\delta_{Ex}$), box-top mixing ratio ($\delta_{Top}$), box-top height ($\delta_{BH}$) and
uncertainties due to data deconvolution as shown in the main text ($\delta_{\text{Deconv}}$).

| | $CH_4$ (%) | $CO_2$ (%) |
|---|---|---|
| $\delta_{\text{M}}$ | 1 | 1 |
| $\delta_{\text{Wind}}$ | 2 | 2 |
| $\delta_{\text{Ex}}$ | 2 | 6 |
| $\delta_{\text{Top}}$ | 8 | 16 |
| $\delta_{\text{BH}}$ | 8 | 3 |
| $\delta_{\text{Deconv}}$ | 1 | 1 |
| $\delta$ | 12 | 18 |

**5.4 Comparison with Gaussian Inversion Approach**

The TERRA computation results can be further evaluated. Of the multiple $CH_4$ plumes that were captured on the north and east walls of the virtual box, the largest $CH_4$ one resembles a nearly perfect Gaussian plume distribution and is clearly associated with the east stack of the two, for which the emission rate may be recalculated using the Gaussian plume model. The Gaussian plume model makes basic assumptions that the plume is emitted from a point source and that the atmospheric turbulence is constant in space and time (Visscher, 2014). In this study, the captured plume was completely elevated and thus not constrained by boundaries. In the absence of boundaries, the equation for pollutant mixing ratios in Gaussian plumes is as follows:

$$c = \frac{Q}{2\pi\bar{u}\sigma_y\sigma_z}exp\left(-\frac{y^2}{2\sigma_y^2}\right)exp\left(-\frac{(z-h)^2}{2\sigma_z^2}\right), \tag{21}$$

where $c$ is the concentration at a given position $x$, $y$ and $z$ (g m$^{-3}$), $Q$ is the emission rate (g s$^{-1}$), $\bar{u}$ is the mean wind speed (m s$^{-1}$), $h$ is the effective source height (m) and $\sigma_y$ and $\sigma_z$ are dispersion parameters in the horizontal (lateral) and vertical directions respectively (m).

The dispersion parameters $\sigma_y$ and $\sigma_z$ were obtained by fitting the spatial distribution of $CH_4$ mixing ratios on the measurement screen into a Gaussian function. As the wall intercepting the plume is not perpendicular to the wind direction, the plume was projected to a different virtual wall perpendicular to the wind direction before fitting the Gaussian function. By calculating the standard deviations of the Gaussian distributions in the y and z directions, $\sigma_z$ is estimated to be 6.3 $\pm$ 0.3 m and $\sigma_y$ is 15.7 $\pm$ 0.4 m. The downwind measurement plane is examined to find the point with the highest $CH_4$ mixing ratio of 6.575 ppm and its location ($s$ = 160 m, $z$ = 217 m). For the separate $CH_4$ plume, the Gaussian plume model gives an emission rate of 40 $\pm$ 6.8 kg h$^{-1}$. The uncertainty is quantified by considering the accuracy of mixing ratio measurement, the variation of wind speed and the confidence interval for the dispersion parameters given by Gaussian function fitting. $CH_4$ measurement uncertainties from the instrument is <1%. The uncertainty contributed by the mean wind speed estimation was examined by varying the average wind speed by the standard deviation of the wind data around the plume (3.8$\pm$0.6 m s$^{-1}$), followed by input into gaussian plume model. This mean wind speed sensitivity analysis resulted in $CH_4$ emission rates that varied by 16%. The same sensitivity analysis was done with $\sigma_y$ (15.7 $\pm$ 0.4 m) and $\sigma_z$ (6.3 $\pm$ 0.3 m), which resulted in $CH_4$ emission rates that varied by 4% and 3% respectively. Thus, the total uncertainty is added in quadrature to be 17%. The TERRA algorithm is able to obtain the emission rate for a selected section through a certain area of the screen. For this isolated $CH_4$ plume, the TERRA algorithm computed an emission rate of 65$\pm$8 kg h$^{-1}$, which is comparable to the emission rate estimation from the Gaussian plume model.

**5.5 Validation of UAV-based Emissions and Comparison with IPCC-based Emissions**

Coking process is one of the highest energy-consuming operations during iron and steel production that tends to emit large amounts of $CO_2$ and $CH_4$. According to the Chinese national GHG inventory report, $CO_2$ and $CH_4$ emissions from coke production in iron and steel production processes were calculated using the Tier 1 method in the IPCC Guidelines (Ministry of Ecology and Environment of of China, 2018). In the Tier 1 method, default emission factors for coke production are used to estimate the $CO_2$ and $CH_4$ emissions without considering local variations, respectively,

$$E_{CO_2} = P_{coke} \times EF_{CO_2} \ and \ E_{CH_4} = P_{coke} \times EF_{CH_4} \ , \tag{22}$$

where $E_{CO_2}$ and $E_{CH_4}$ represents the $CO_2$ and $CH_4$ emission rates from coke production, $P_{coke}$ represents coke production, $EF_{CO_2}$ and $EF_{CH_4}$ are the IPCC default emission factors for $CO_2$ and $CH_4$, which are 0.56 t $CO_2$/t of coke and 0.1 g $CH_4$/t of coke, respectively. The measured Shagang coking plant consists of two coke oven batteries, each with its own stack. Each battery produced 127.8 t coke $h^{-1}$, thus totalling 255.6 t coke $h^{-1}$ ($P_{coke}$) between the two batteries during the UAV measurement period with a coke yield of 78.5%. A material balance analysis revealed that $CO_2$ emitted from the stacks during the full coking process was 103±32 t $CO_2$ $h^{-1}$ (SI). In comparison, the UAV measurement-based emission rate obtained in this study is 110±18 t $CO_2$ $h^{-1}$, which is consistent with the $CO_2$ emissions based on the material balance analysis. For comparison, multiplying the IPCC default emission factor with the coke production at the Shagang coking plant yields an emission rate from coking of 143 t $CO_2$ $h^{-1}$, higher than either the material balance based result by about 39% or the UAV-based result by 30%. This suggests that the IPCC default emission factor is too high for this particular coking plant.

On the other hand, the UAV-measurement based emission of 0.12±0.014 t $h^{-1}$ for $CH_4$ is four orders of magnitude higher than $1.28 \times 10^{-5}$ t $h^{-1}$ emissions for $CH_4$ estimated using the IPCC Tier 1 emission factor $EF_{CH_4}$. The IPCC emission factor for coke production is derived by averaging plant-specific $CH_4$ emissions data for 11 European coke plants reported in the IPPC I&S BAT Document (European IPPC Bureau, 2001), but information about the data collection method such as sampling methods, analysis methods, time intervals, computation methods and reference conditions is not available according the report. It is important to note that the present UAV measurement represents a one-time measurement where there was only one flight conducted in this campaign. The result clearly serves the purpose for validating the overall methodology from air sampling and analysis, computing the emission rates, to estimating the associated errors. The fundamental assumption in the mass balance approach is that plumes and emissions remain constant throughout the measurement period. Given the short duration of the flight and the good comparison between the present emission result and the material balance emission estimate, such an assumption appears to be valid. However, a hypothesis of a constant

emission rate over time remains to be tested. Conducting multiple flights over time, computing emission rates and
assessing their uncertainties will allow for statistical sampling of the probability distribution of the emission rates and
hence deriving the mathematical expectation of the emission rate. Only then the derived emission factors can be used for
inventory preparation and/or comparison with existing ones with statistical confidence. Given the limited circumstance
of having only one flight in this study, it becomes clear such purpose cannot be achieved. Consequently, the emission
values of $CH_4$ derived from measurements in this section are only suitable for qualitative comparisons with published
emission factors. The comparison results indicate that real-world emission factors may significantly differ from the default
emission factors but more work is needed. The additional $CH_4$ may come from the leakage of the coke oven gas when it
is recycled as fuel in firing the coke oven (SI). Both reasons point to a need for further emission measurements to
determine the local emission factors and a further validation of the $CH_4$ emission factors of coke production.
**6 Conclusions**
In this paper, we present the development of a UAV measurement system for quantifying GHG emissions at facility levels.
The key element of this system is a newly designed air sampler, consisting of a 150-meter-long thin-walled stainless steel
tube with remote-controlled time stamping. Through laboratory testing, we found that the air sampler generated smoothed
time series data compared to online measurement by the CRDS analyzer. To addressing the smoothing effect, we
developed a deconvolution algorithm to restore the resolution of the time series obtained by the air sampler. For field
validation, the new UAV measurement system was deployed at the Shagang Steel to obtain $CO_2$ and $CH_4$ emissions from
the main coking plant at Shagang Steel. Mixing ratios of $CO_2$ and $CH_4$ together with meteorological parameters were
measured during the test flight. The mass-balance algorithm TERRA was used to estimate the coking plant $CO_2$ and $CH_4$
emission rates based on the UAV-measured data. For further analysis, we compared these emission results with those
derived using Gaussian plume inversion approach and carbon material balance methods, demonstrating good consistency
among different approaches. In addition, when compared the top-down UAV-based measurement results to that derived
from the bottom-up emission inventory method, the present findings indicated that the IPCC emission factors can be
significantly different from the actual emission factors..
*Acknowledgment.* This project was supported by a grant from the National Natural Science Foundation of China Creative
Research Group Fund (22221004).
*Data availability.*  Data are available upon request by the corresponding author.
*Author contribution.*  TH, CX, YL and, SML conducted the fieldwork with the support by XG, XZ, and FB. TH and CX
conducted laboratory experiments with the guidance by SML. TH performed the primary data analysis, and wrote the
initial draft of the manuscript. YH provided expertise in model analysis. Algorithm programming was provided by YL.
YY and YZ did the wind data correction. SML reviewd and edited the manuscript, and ensured the accuracy and integrity
of the study.
*Competing interests.*  The authors declare that they have no conflictof interest.

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
