# Peer review of "Development of a continuous UAV-mounted air sampler and"

_Atmospheric Measurement Techniques, 2023_

## Author Response (AR1)

**Response to reviewers**

Dear editors and reviewers:

Thank you for taking the time to review our paper. We appreciate your valuable feedback and constructive comments, which have greatly contributed to improving the quality of our work. We have carefully considered each of your comments and suggestions, and we are pleased to provide our responses below (Our responses are in blue, and our edits to the manuscript are in red):

**Referee #1**

\# General comments

This is a well-structured, well-contextualised piece of work that has the potential to be a valuable addition to the literature. Well done! It focuses on an application of the AirCore concept with custom-built equipment. In test surveys on a coking plant, $CH_4$ emissions of several orders of magnitude larger than inventory estimates are reported.

However, the paper needs work in order to make it reproducible and needs clarification about the use and modification of algorithms and reporting of uncertainty. In addition, the wind measurements appear to give much more consistent results than most papers I have read on this subject - it would be good to see some detail on how this was achieved.

We have moved "**Section SI-1. TERRA-based determination of $CH_4$ and $CO_2$ emission rates**" and "**Section SI-2. Uncertainty estimation**" from Supporting Information to the main text, please see **Section 2.5 Mass balance approaches for determining emission rates** and **Section 5.3 Uncertainty Analysis** in the revised manuscript. In addition, we have added more details to the use and modification of the algorithm and reporting of uncertainty, including:

1) We have added the parameterization of the Kriging interpolation.

   "The kriging weights were obtained with an isotropic spherical semivariogram model. In TERRA, nugget, sill, and range can all be modified to fit the semivariogram model."
   "The semivariogram of the flight points was fitted with a spherical model (range=300, sill=3, nugget=0)."

2) We have added the equations for each term of emission rate estimation in TERRA ($E_{C,H}$, $E_{C,V}$, $E_{C,M}$) to clarify the algorithm.

   To summarize, in TERRA the mass-balance in computing the emissions within a control box for a given inert pollutant such as $CH_4$ or $CO_2$ is presented by:

   $$E_C = E_{C,H} + E_{C,V} - E_{C,M}$$

where $E_C$ is the emission rate, $E_{C,H}$ is the horizontal advective transfer rate through the box walls, $E_{C,V}$ is the advective transfer rate through the box top and $E_{C,M}$ is the increase in mass within the volume due to a change in air density. Other terms listed in the Gordon et al. computation algorithm that were used to solve for the total emission rate were often neglected as they contribute little to the total emission rates (Gordon et al., 2015). Each term is estimated as:

$$E_{C,H} = M_R \iint X_C \rho_{air} U_\perp ds dz$$

$$E_{C,V} = M_R X_{C,Top} \iint \rho_{air} \omega dx dz$$

$$E_{C,M} = M_R \iiint X_C \frac{d\rho_{air}}{dt} dx dy dz$$

where $M_R$ is the ratio of the compound molar mass to the molar mass of air, $X_C(s,z)$ is the mixing ratio of the compound in question, $\rho_{air}(s,z)$ is the air density, $\omega$ is the vertical wind velocity at the box top, $X_{C,Top}$ is the mixing ratio at the top of the box, and $U_\perp(s,z)$ is the horizontal normal wind vector to the flight track calculated from the northely and easterly components $(U_E(s,z), U_N(s,z))$:

$$U_\perp = \frac{\frac{U_E \partial s}{\partial x} - \frac{U_N \partial s}{\partial y}}{\sqrt{(\partial s/\partial x)^2 + (\partial s/\partial y)^2}}$$

The vertical transfer rate term $E_{C,V}$ is estimated by computing the air mass vertical transfer rate, determined from vertical wind estimated from air mass balance within the box, and multiplying it with the $CO_2$ or $CH_4$ mixing ratios at the box top. This term is normally negligible in other top-down emission estimate approaches since it is typically miniscule compared to horizontal fluxes, but can affect the computed emission rates when vertical air movement becomes more significant such as under unstable atmospheric conditions. $E_{C,M}$ is often ignored in other mass-balance approaches; in TERRA it is estimated by taking the time derivative of the ideal gas law in temperature and pressure during the flight time, and typically it does not change significantly over the duration of 30 minutes or so for the UAV flight.

3) We have added a paragraph to introduce the modification of TERRA to suit the UAV measurements:

"To suit the UAV measurements, the following modifications to the TERRA algorithm were made: (1) A much higher interpolation resolution for the kriging mesh was implemented for application to the UAV measurements in this study, with the interpolation mesh size adjusted to 1 m (vertical) by 2 m (horizontal), as UAVs fly significantly shorter distances compared to applications to piloted aircraft for which the interpolation resolution was 20 m (vertical) by 40 m (horizontal); (2) The modified TERRA now applies an embedded routine to automatically fit flight tracks using least squares, while this procedure was previously conducted manually offline through geographic information system when using TERRA. (3) The modified version of TERRA has added an algorithm for correcting negative weights during Kriging interpolation

following Deutsch (1995)."

4) We have added more calculation processes and details for deriving each term of uncertainty.

"The accuracy of the mixing ratio measurements from the Picarro CRDS analyzer is 50 ppb and 1 ppb for $CO_2$ and $CH_4$, respectively. By adding variations in the measured mixing ratios based on the measurement accuracies and re-applying TERRA, the derived emission rates varied within 1% for both $CO_2$ and $CH_4$. Thus, the uncertainties in the emission rates due to mixing ratio measurements ($\delta_M$) were estimated at 1% for both $CH_4$ and $CO_2$.

The anemometer measures wind speeds with an accuracy of $\pm 0.1$ m s$^{-1}$ at wind speeds <10 m/s and wind directions with an accuracy of $\pm 1°$. The uncertainty of the wind measurements ($\delta_{Wind}$) was estimated using error propagation in the normal wind $U_\perp$(s,z), as it is calculated from the northerly and easterly wind components, thus from wind speed and wind direction:

$$\delta_{U_\perp} = \sqrt{\delta_{easterly}^2 + \delta_{northly}^2 + 2\sigma_{easterly-northly}}$$
$$\delta_{easterly} = |WScos(WD)\sigma_{WD}|$$
$$\delta_{northly} = |WSsin(WD)\sigma_{WD}|$$

Using this calculation, the uncertainty of the normal wind $\delta_{U_\perp}(s,z)$ was derived at each location. The uncertainty contributed to the total emission rates to the overall computed emission rate was examined by setting the normal wind to its upper and lower bounds defined by its uncertainty range, followed by computing the emission rates using TERRA. The derived $CH_4$ and $CO_2$ emission rates varied by 1.5% and 1.9% respectively. Hence the uncertainties from wind speed measurements ($\delta_{Wind}$) were conservatively estimated to be 2% for both $CH_4$ and $CO_2$.

Due to a lack of near-surface measurements along the box walls, extrapolation of $CH_4$ and mixing ratios from the lowest flight path (~ 150 m above ground level) to the ground level has been shown to be a source of potentially large uncertainty within TERRA. The magnitude of the uncertainty depends on the nature of the emissions; for example, surface emissions which may not be fully captured by the flight altitude range have higher uncertainties at ≈20%, whereas elevated stack emissions which are fully captured by flight altitude range lead to much smaller uncertainties of <4% in the emission estimates (Gordon et al., 2015). In the present study, to estimate uncertainties due to extrapolating mixing ratios from the lowest flight track to the ground ($\delta_{Ex}$), results from all extrapolation techniques (i.e., linear to the ground, constant value to the ground, linear to background value, or some combination of methods) were derived and compared with the result using extrapolation from the lowest flight to background levels. Therefore, this term of uncertainty was evaluated at 2% and 6% for $CH_4$ and $CO_2$ respectively.

Table1. Emission rates derived using different extrapolation techniques

| Extrapolation techniques | All background below flight altitude (this | Constant value from lowest flight altitude to | Linear between constant and background at | linear | exponential |
|---|---|---|---|---|---|

| | study) | surface | surface | | |
|---|---|---|---|---|---|
| CH$_4$ Emissions(kg/hr) | 115.7 | 113.9 | 116.9 | 113.9 | 113.6 |
| CO$_2$ Emissions(kg/hr) | 110100 | 109970 | 109400 | 109970 | 103960 |

Additional components contributing to uncertainties in the computed emission rates specific to the box approach include box-top mixing ratio ($\delta_{Top}$) and box-top height ($\delta_{BH}$). The TERRA box approach assumes a constant mixing ratio at the box top ($X_{C,Top}$) by averaging the measured value at the top level. The term $\delta_{Top}$ is determined from the 95% confidence interval ($2\sigma/\sqrt{n}$) of the interpolated measurements. The calculated confidence interval of the mixing ratio at the box top is 0.01 ± 0.13 ppm for CH$_4$ and 70.1 ±89.1 ppm for CO$_2$. A top average mixing ratio of 0.14 ppm for CH$_4$ and 159.2 ppm for CO$_2$ are set as input parameters to derive resulting uncertainties in the emissions rates. Thus, 106.6 kg/hr for CH$_4$ and 93760 kg/hr for CO$_2$ were derived. Thus, this uncertainty term is conservatively taken as 8% and 16% for CH$_4$ and CO$_2$.

The uncertainty due to the choice of box height, $\delta_{BH}$, within TERRA is estimated by recomputing the emission rate with a reduced box height (z) of 100 m. The recalculated emission rate after reducing the box height of 100m is 106.4 kg/hr for CH$_4$ and 113500 kg/hr for CO$_2$, thus $\delta_{BH}$ is estimated as 8% for CH$_4$ and 3% for CO$_2$.

For cases that use the air sampling system instead of online measuring instruments, as the CH$_4$ and CO$_2$ time series measured from the air sampler were deconvoluted to restore the unsmoothed time series before being input into the TERRA algorithm, it is necessary to account for the uncertainty that comes from such deconvolution as outlined in the main text. Time series before and after deconvolution were applied to the TERRA algorithm to obtain the total emission rates. The computations show that emission rates before and after deconvolution vary within 1%, which was taken as the uncertainty $\delta_{deconv}$. The assessment of uncertainties for the TERRA-computed emission rates from the coking plant are listed in Table 2."

Table 2. Assessment of percent uncertainties for CH$_4$ and CO$_2$ emission rate estimations

| | CH$_4$ (%) | CO$_2$ (%) |
|---|---|---|
| $\delta_M$ | 1 | 1 |
| $\delta_{Wind}$ | 2 | 2 |
| $\delta_{Ex}$ | 2 | 6 |
| $\delta_{Top}$ | 8 | 16 |
| $\delta_{BH}$ | 8 | 3 |
| $\delta_{Deconv}$ | 1 | 1 |
| $\delta$ | 12 | 18 |

In addition, more details of the post-flight correction for wind speed and direction have been added to the main text.

One important note is that the results of the paper relies on work from Yanrong Yang on windspeed correction - this appears to be groundbreaking work and is "in preparation" but needs to be available to reviewers in order for the paper to pass review according to AMT submission rules - I would be grateful if the authors can upload this.

Yes we also believe the work described in Yang et al. paper is new for its kind. It is ready for submission. We have uploaded the manuscript on wind speed correction for your reference.

The section on deconvolution is impressively well-written, and I would like to see the rest of the paper up to that standard. More reference should be made to the supplementary material where appropriate - the material mass balance in supplement section 3 is also well written, and another good benchmark for what should be added to the rest of the paper. However, supplementary material critical to understanding and replication of the paper should be moved to the main body of the text, particularly regarding uncertainty and data processing methods.

Thanks for your advice! The TERRA methods and uncertainty analysis has been moved to the main text. The material mass balance was kept in the supplementary information in consideration of the length of the manuscript.

**Specific comments**

The paper goes into a good deal of depth on deconvolution of smoothed mixing ratios of methane and CO2. However, the main impression I have is that this depth is missing in some other critical areas of the analysis, such as how the TERRA algorithm was applied, how modifications to it were made and how its various assumptions and parameters were defined. There is some detail in the supplementary material but more is needed. More detail on the exact methods of fitting the gaussian model would also be beneficial. If someone else was to replicate this paper, they should have all of the detail needed in order to do so. I was also curious as to why TERRA was chosen as to my knowledge it is not widely used in UAV emissions measurements

We have added more descriptions of how TERRA was applied, including the parameter optimization of Kriging, a clearer presentation of the equations, and the modifications to the algorithm as addressed above.

We have also added more details on the Gaussian Inversion Model:

"By calculating the standard deviations of the Gaussian distributions in the y and z directions, $\sigma_z$ is estimated to be 6.3±0.3m and $\sigma_y$ is 15.7±0.4m. The downwind measurement plane is examined to find the point with the highest $CH_4$ mixing ratio of 6.575 ppm and its location (s=160m, z=217m)."

With regard to the choice of TERRA for computing emissions, in previous work one of us (S.-M.

Li) has validated the method using piloted aircraft measurements (Gordon et al., 2015), and we believe that this is a rigorous and theoretically sound method to make UAV measurements for emission estimation because it relies on in-situ measurements of both 3-d wind and met parameters and concentration data and achieves true mass balance within the virtual flight box. Indeed, the comparison between the TERRA computed results and the carbon material balance gives us confidence that the methodology (measurements + emission computing algorithm) works well. Importantly, we are able to achieve this result because we have good concurrent in-situ wind data after using a wind correction algorithm (Yang et al. paper, soon to be submitted). To achieve its rigor, TERRA needs both accurate concentration data (mixing ratio data) and accurate wind data to achieve mass balance and derive reliable emission results. We believe that other UAV-based emission measurements need to consider the various (inherent) assumptions, such as suitability of wind data or the degree of atmospheric mixing, and report the uncertainties therein in computing and reporting their emissions.

Regarding the title - I wasn't clear about what in the measurement methodology is actually new given that the AIrcore concept is well-demonstrated; I would suggest "Development of a continuous UAV-mounted air sampler and application to the quantification of CO2 and CH4 emissions from a major coking plant" or alternatively a better explanation of what is new in the methodology. Even if the methods are well-established, the paper is a good one and a helpful aid for applied science.

Thanks for your advice! This title is indeed better and we are pleased to adopt it as our title.

While the AirCore concept is not new, the overall methodology is new;

It is suggested that methane emissions measured are anomalous - by several orders of magnitude - compared to emissions factor-based inventory reporting. This is a plausible, interesting and important result, if true. However, it seems that this may - and please make this more clear if I have misunderstood - be based on only one measurement flight. Measurement uncertainty is also presented in a way that could lead the reader to assume that it gives an idea of the error (i.e. accuracy) rather than the precision of an individual measurement.

Although there are thousands of mixing ratios measured and various other components to measurement uncertainty (e.g. uncertainty around wind speed, and hopefully around wind direction although this is not clear), the basic unit of measurement is the flight. Random error will be potential very high when doing atmospheric measurements of this kind - plumes move and change shape in ways that are difficult to account for even within one measurement. Averaging of multiple flights and their uncertainties will allow for quantification of this error.

If there was only one flight conducted in this campaign, then it represents a pilot study, and will need replication and quantification in order to draw conclusions about emissions inventories. Unfortunately it is difficult to make conclusions about any one measurement when we have little idea about the spread. If there were more, then this should be better explained in the text.

It is important to note that the paper is primarily a method validation paper, as such one flight was

sufficient for the comparison purpose. But yes there was only one flight conducted in this campaign and we agree with your concern regarding the representativeness of the data for comparison with published emission factors. Thus we have added a few sentences to the text to make the discussion sound more conservative:

"It is important to note that the present UAV measurement represents a one-time measurement where there was only one flight conducted in this campaign. The result clearly serves the purpose for validating the overall methodology from air sampling and analysis, computing the emission rates, to estimating the associated errors. The fundamental assumption in the mass balance approach is that plumes and emissions remain constant throughout the measurement period. Given the short duration of the flight and the good comparison between the present emission result and the material balance emission estimate, such an assumption appears to be valid. However, a hypothesis of a constant emission rate over time remains to be tested. Conducting multiple flights over time, computing emission rates and assessing their uncertainties will allow for statistical sampling of the probability distribution of the emission rates and hence deriving the mathematical expectation of the emission rate. Only then the derived emission factors can be used for inventory preparation and/or comparison with existing ones with statistical confidence. Given the limited circumstance of having only one flight in this study, it becomes clear such purpose cannot be achieved. Consequently, the emission values of $CH_4$ derived from measurements in this section are only suitable for qualitative comparisons with those used published emission factors. The comparison results indicate that real-world emission factors may significantly differ from the default emission factors but more work is needed."

In general I would like to see a much more robust approach to quantifying uncertainty and confidence intervals.

See the revisions as listed above.

**Line-by-line**

Line 57-59 - "could" implies that UAVs are not current used - they are widely deployed in commercial operations to quantify methane emissions flux already. This goes back to the question over the title.

Thanks for the suggestion. We have deleted "could" in this sentence:

"Most importantly, a UAV platform fills the sampling space between ground and altitudes of up to hundreds of meters above ground, in which other mobile platforms have been unable to operate(Shaw et al., 2021). Due to their relatively low flying speeds, UAV platforms offer a high spatiotemporal resolution for sampling and thus enabling accurate plume mapping."

Line 141 - Post-processing here is quite casually stated. I would like to see some reference to the detail of Yang's approach here - this itself is innovative and worth going into more detail, but it's also important to clearly describe the methods even if the full detail is elsewhere. The method relies

fairly heavily on this correction and AMT requires "in preparation" manuscripts to be available to reviewers, so I think we need to see this paper to be able to assess this part of the method. Windspeed measurement on drones is tricky to get right, and I think the method has to be made more explicit to give the results credence.

Yang's paper is uploaded for reference. A brief detail on Yang's approach to wind data correction has been added to the main text:

"The anemometer in this research is mounted at an upward distance of 70 cm from the center of gravity of the UAV. A full digital model of the UAV, the anemometer and its mounting frame, and the air sampler was created. Using this digital model, computational fluid dynamics (CFD) simulations were performed to quantify wind speed disturbances caused by the UAV's rotor propellers on the anemometer during flight under a vast array of different wind conditions. An overall correction algorithm was developed in which parameters for propeller disturbances determined based on the CFD simulations were included along with correction schemes for false signals resulting from translational motions and changes in UAV pitch, roll and yaw. The correction algorithm was verified with real-world UAV flight-meteorological tower measurement intercomparisons (Yang et al., 2023)."

Line 108 /113- the CO2 marker - I was unclear on why this is necessary - is it not possible to map sensed molar ratios to GPS locations without this marker? I assume it's an important part of your technique but I'd like to know more about why

The air samples in the tubing are analyzed with the Picarro CRDS analyzer after UAV flight, thus we need to know when the sampling begins to identify the starting point of the time series. The $CO_2$ markers result from pulses of air with low/zero $CO_2$ concentrations, producing low deviations in the Picarro instrument output from the signals for the background air $CO_2$ concentrations. $CO_2$ markers are set at the beginning of every flight and at specifically set times, so that they produce very evident negative signals in the $CO_2$ time series. These markers allow us to match the starting time of the sampling to the data point of the first $CO_2$ marker and later the other times when the markers are known to be set. We have added the following sentence to the main text:

"The $CO_2$ marker helps to identify the starting point of the UAV air sampling during later analysis."

Line 112 - are there disadvantages to waiting longer, e.g. if flights were longer? Is there a difference between samples at the start of the flight that are e.g. 30 minutes old, vs those at the end of the flight?

Limited by the battery capacity of the UAV, a flight usually lasted within 30 minutes, a duration within which the smoothing effect remains acceptable. We have added the following sentence to the main text:

"Waiting longer would lead to unwanted mixing of the samples in the tubing. The air sample enter the tubing from the air inlet during sampling and leave the tubing from a different air outlet during later analysis. As a result, the samples at the beginning of the flight spend the same amount of time within the tubing as those at the end of the flight."

Line 150 - it is quite uncommon for vertical precision to be better than horizontal for GPS - are you sure this is correct?

Manufacturer specification reports that the horizontal hovering precision of the GPS on the UAV is ± 2 m and the vertical hovering precision is ±1.5 m. We have added "hovering" to the original sentence.

Line 155 - can you clarify the discussion of why laminar flow would lead to smoothing of concentration changes?

This sentence was improperly written. We have revised it as follows:
"However, direct withdrawal of air from the sample tubing by the analyzer at a flow rate as low as the sampling flow rate of 18 sccm results in smoothing of concentrations from the inner wall surface drag and desorption inside the tubing"

Line 275 - approximately how long was each flight? Do you expect the plume to have systematically moved during flight, and how do you account for this?

The flight lasted for approximately 30 minutes. It's assumed that the plume remains steady during the time of measurement.

Line 294 - please provide confidence intervals or similar for windspeed and direction. Did these vary at all according to the direction of travel of the drone? Measuring windspeed on drones is notoriously difficult and you appear to have extremely consistent results compared to most other datasets I have seen. Reviewers would need to see the paper in preparation that these results rely on.

Agreed, wind measurements from UAVs are difficult. After applying the correction algorithm for wind measurements, the wind direction and speed remain very similar regardless of the direction of the UAV flight. This is in fact a standard wind calibration protocol for wind measurements from aircraft. Such results are given in Figure 5. Also see the response and revisions regarding wind measurements from the UAV above. The Yang et al. paper is ready for submission. As stated in the Yang et al. paper, we believe that our work on wind correction algorithm for measurements made from UAVs is the first of its kind.

We have added the standard deviation for windspeed and wind direction:

"The average windspeed is 4.7±4.9 m/s and the average winddirection is 216.4±38.4° during the time of flight."

Basically wind speed and direction did not vary according to the direction of travel of the drone during the time of measurement:

[Figure]

Wind measurement at each point on each side of the flight box

We have also uploaded the manuscript of Yang's work on wind speed correction for your reference.

Line 300 - how did you handle negative deconvolved measurements, such as those visible on the CO2 graph?

The negative values visible on the $CO_2$ graph are $CO_2$ markers. However overshooting did exist during deconvolution, we replace the overshooting values by the background values as shown in the graph below (the top graph is before removing overshooting values and the graph below is after removing overshooting values)

[Figure]

Line 306 - Can you be specific about the TERRA algorithm, your modifications to the algorithm and how it is applied? Kriging of gases in air is a complex business and the use of various kriging parameters, semivariogram optimisation, anisotropic search etc. needs to be clearly stated and

reasons given for each parameter choice. I see some information in the supplement, but not enough to replicate your method I think.

[Figure]

Before Kriging, each flight position point from a three-dimensional position of latitude (y), longitude (x), and altitude (z, above mean sea-level) is translated to a two-dimensional screen position of horizontal path distance s = f (x,y), and altitude, z. The kriging weights were obtained with an isotropic spherical semivariogram model. In TERRA, nugget, sill, and range can all be modified to fit the semivariogram model. Kring resolution can also be adjusted depending on the flying speed of the UAVs or airplanes. The figure above shows the GUI of Kriging in TERRA, the semivariogram of the flight points at the coking plant are shown with a spherical semivariogram model fitted to it (range=300, sill=3, nugget=2, horizontal resolution=2m, vertical resolution=1m). In addition, negative weights arised during kriging were corrected using the algorithm proposed by Deutsch et al.

We have added the details of Kriging to the main text of the manuscript:

"The kriging weights were obtained with an isotropic spherical semivariogram model. In TERRA, nugget, sill, and range can all be modified to fit the semivariogram model."
"The semivariogram of the flight points was fitted with a spherical model (range=300, sill=3, nugget=0)."

Line 308 - The distance between horizontal lines is 15m, yet the kriging nodes are 2m in the vertical. How did you decide on this resolution? I see in the supplement - please refer to it more often! - that there was a 20x reduction from aircraft resolution. Why this magnitude and not more/less?

The kriging resolutions in this study are determined based on the sampling resolutions, which, in turn, are influenced by the speed of the vehicles being used. For instance, unmanned aerial vehicles (UAVs) typically fly at a speed of approximately 10 m/s, whereas airplanes generally fly at a much faster speed of around 200 m/s. Consequently, the sampling resolution for UAVs is approximately 10 m, while for airplanes, it is about 200 m. The kriging resolutions in this study are not quantitative results. Typically, interpolation resolutions are set to be higher than the sampling resolutions; however, setting them excessively high can result in costly calculations. Therefore, a balanced approach is taken, with the interpolation resolutions set in the middle. Modifying the interpolation resolution within a reasonable range has minimal impact on the final outcome of the study.

Line 315 - can you be specific and detailed about how uncertainty was obtained here - I see in the supplement (please refer to it) how you combine them but not how they were actually calculated except through a reference to Gordon et al.

Was wind direction not an uncertainty component? This would likely be non-symmetrical too, as it varies by the cosine of the angle between the wind and the measurement plane. How does TERRA deal with the case where there are two walls that intersect the plume? Is there an implicit assumption that all points in the measurement plane are equidistant from the source(s)? If not, how is that accounted for?

Details on how to derive each term of uncertainty have been added to the main text in **Section 5.3 Uncertainty Analysis**.

In TERRA, the wind vector is decomposed into two parts, one normal to the flight track and the other parallel, when estimating the mass flux of a pollutant. A flux at each location is computed by mixing ratio $X_C$(s,z), air density $\rho_{air}$ (s,z), and the normal wind speed $U_\perp$(s,z). The wind speeds, after applying the wind correction algorithm to the measurement data, at each location are separated into northerly and easterly components $(U_E(s,z), U_N(s,z))$. Then, the normal wind vector $(U_\perp$(s,z), positive outwards) is calculated as

$$U_\perp(s,z) = \frac{U_E(s,z)\partial s/\partial x - U_N(s,z)\partial s/\partial y}{\sqrt{(\partial s/\partial x)^2 + (\partial s/\partial y)^2}}$$

Line 323 - how were delta-mixing ratios arrived at, i.e. what was your baselining procedure?

Background $CH_4$ and $CO_2$ were determined using upwind measurements. The background between upwind data was linearly interpolated and box-car smoothed within a 3-4 minute moving window to derive a variable baseline $CH_4$ and $CO_2$ for the entire 30-minute flight.
We have also added this sentence to the manuscript.

Line 323 - It seems plausible given Figure 5 that the plume moved during measurements, and that you may have double-counted it. How is this reflected in your results? Are you confident that there was no double counting? Would this method work on a day that had less consistent wind direction over the 30 minute measurement period? Were there any trends in windspeed during the measurement period and how was this controlled for?

According to the staff at the coking plant, the two coking stacks are both running continuously during the period of measurement. Furthermore, the wind measurements indicated a consistent wind field during the entire flight. Therefore we assume the two plumes come from different stacks rather than double-counting. The material balance analysis in SI revealed that $CO_2$ emitted from the stacks during the full coking process was $103\pm32$ t $CO_2$ h$^{-1}$, while the emissions of the two plumes are 64 t $CO_2$ h$^{-1}$ and 38 t $CO_2$ h$^{-1}$ respectively.

Uncertainties increase when there is a lack of consistent wind. Thus, we usually choose to conduct

the UAV flight campaign where there are suitable meteorological conditions to obtain more accurate results.

We did not average over the entire set of measurements for windspeed, instead each concentration measurement is combined with paired windspeed to calculate a flux. The graphs below reveal the trend of wind speed and wind direction during the time of measurement:

[Figure]

Line 330 - does this not apply to the time-averaged gaussian plume, rather than the instantaneous plume that you measured, which should be somewhat non-gaussian or at least with a smaller sigma and higher maximum concentration? How are you sure that you correctly paramaterised the maximum, i.e. found the highest concentration value? Were there differences between sigma y and sigma z?

The flight time lasted for 30 minutes, and we assume that the plume remained steady and consistent during the time of measurement. Therefore the plume measured is assumed to be time-averaged Gaussian plume rather than an instantaneous plume. We examined the measurement plane and found the highest concentration value of 6.575 ppm and its location (s=160m, z=217m). Sigma z is 6.3 and sigma y is 15.7.

Line 341 - how was this confidence interval arrived at. Are you using normally distributed uncertainties or something else?

CH$_4$ measurement uncertainties from the instrument is <1%. The uncertainty contributed by the mean wind speed estimation was examined by varying the average wind speed by the standard deviation of the wind data around the plume(3.8±0.6 m/s), followed by input into gaussian plume model. This mean wind speed sensitivity analysis resulted in CH$_4$ emission rates that varied by 16%. The same sensitivity analysis was done with sigma y (6.3±0.3 m) and sigma z (15.7±0.4 m), which resulted in CH$_4$ emission rates that varied by 4% and 3% respectively. Thus, the total uncertainty is added in quadrature to be 17%

The uncertainty analysis of the Gaussian Inversion Approach has been added to the main text.

Line 344 - I would like to see TERRA measurement uncertainty reported in the main text by its components and how they were arrived at (normally distributed uncertainty, potential range analysis etc.)

More details of uncertainty analysis has been added to the main text. Please see **Section 2.5 Mass balance approaches for determining emission rates**

Line 345 - I would not go so far as to say that two models agreeing with each other means that either one is reliable - that's what your validation should do.

Thanks for pointing this out, we have removed this sentence.

Line 358 - was the material balance analysis from daily process data? Where did it come from? please refer to the supplementary material as appropriate - I see there is a detailed section there. Was the production rate constant? Any process variations during the period of measurement? Your measurement took quite a long time I think - do you have any comment on what the maximum length of time for a suitable measurement, given there can be process variations within its timeframe?

According to the staff at the coking plant, the coking batteries are in continuous operation and the coking time (the time from charging the coal to pushing of coke out of the oven) is 26 hours. Thus a 30-minute flight is a suitable length of time for measurement and material balance analysis.

We have also added this information to the SI.

Line 372 - please refer to the supplement here so I know to look there for plant information - I'm not entirely clear what taps leakage or door leakage would be even after reading it though

We have edited this sentence to "The additional CH$_4$ may come from the leakage of the coke oven gas when it is recycled as fuel in firing the coke oven (SI)".

Supplement - Line 69 - These deltas appear to be based on the Monte Carlo from Gordon et al. but those were specific to their own measurements, and probably quite dependent on the number of measured points. To illustrate - if you have few signal points, wind speed variation can have a big impact, but in big diffuse plumes like in Gordon et al it will make much less difference. You should consider your own Monte Carlo, or alternatively use normally-distributed uncertainties and accept their more conservative results.

I would really like to see more detail on uncertainty propagation here and in the main text. Wind direction is not included here - this can be a large source of potential systematic bias when a plane does not intersect a plume at right angles, and thus biases measurements high according to a cosine relationship. It doesn't seem like Gordon et al go into this either, to be fair.

We now calculate uncertainties from wind speed and mixing ratiomeasurements the as the following method:
The accuracy of the mixing ratio measurements from the Picarro CRDS analyzer is 50 ppb and 1 ppb for $CO_2$ and $CH_4$, respectively. By adding variations in the measured mixing ratios based on the measurement accuracies and re-applying TERRA, the derived emission rates varied within 1% for both $CO_2$ and $CH_4$. Thus, the uncertainties in the emission rates due to mixing ratio measurements ($\delta_M$) were estimated at 1% for both $CH_4$ and $CO_2$.

The anemometer measures wind speeds with an accuracy of $\pm0.1$ m s$^{-1}$ at wind speeds <10 m/s and wind directions with an accuracy of $\pm1°$. The uncertainty of the wind measurements ($\delta_{Wind}$) was estimated using error propagation in the normal wind $U_\perp(s,z)$, as it is calculated from the northerly and easterly wind components, thus from wind speed and wind direction:

$$\delta_{U_\perp} = \sqrt{\delta_{easterly}^2 + \delta_{northly}^2 + 2\sigma_{easterly-northly}}$$
$$\delta_{easterly} = |WScos(WD)\sigma_{WD}|$$
$$\delta_{northly} = |WSsin(WD)\sigma_{WD}|$$

Using this calculation, the uncertainty of the normal wind $\delta_{U_\perp}(s,z)$ was derived at each location. The uncertainty contributed to the total emission rates to the overall computed emission rate was examined by setting the normal wind to its upper and lower bounds defined by its uncertainty range, followed by computing the emission rates using TERRA. The derived $CH_4$ and $CO_2$ emission rates varied by 1.5% and 1.9% respectively. Hence the uncertainties from wind speed measurements ($\delta_{Wind}$) were conservatively estimated to be 2% for both $CH_4$ and $CO_2$.

More details on uncertainty propagation have been added to the main text in Section 5.3 Uncertainty Analysis. As addressed above, wind direction is considered when deriving the mass flux of a pollutant, the normal wind vector (perpendicular to the flight track of the UAV) is calculated at each measured location.

With regard to the plane not intercepting the plume at the right angles, this issue is taken care of in the TERRA through the computation of the normal wind vector to the flight path, in which the

normal wind vector ($U_\perp$(s,z), positive outwards) is calculated as

$$U_\perp(s,z) = \frac{U_E(s,z)\partial s/\partial x - U_N(s,z)\partial s/\partial y}{\sqrt{(\partial s/\partial x)^2 + (\partial s/\partial y)^2}}$$

Thus your concern about the cosine factor has been already considered in the TERRA computation. Gordon et al. (2015) had in fact given a detailed explanation of how this normal wind vector was computed.

You use delta uncertainties, which is fine, but these can be nebulous unless well-defined - I would like to understand each one and where it comes from, so that the paper is replicable. The relationship of some of these to the calculation are not clear to me (e.g. box-top mixing ratio, box-top height etc.)

We have explained how each term of uncertainty is calculated in Section 5.3 Uncertainty Analysis.

Supplement - Line 77 - This seems highly surprising to me, regarding wind speed. A very accurate and precise anemometer might have a 1% 1 sigma error but surely you are averaging over the entire set of measurements for windspeed? Or does each concentration measurement combine with paired windspeed to arrive at a flux? If that is true, it implies significant assumptions about dispersion of the gas between the source and the point of measurement and these need clearly stated. For example - do you assume that a measurement taken in a gust has correspondingly lower mixing ratios of the target gas, and thus a similar flux to the same situation at a lower windspeed with correspondingly higher mixing ratios?

In TERRA, each concentration is paired with concurrent and co-located windspeed to determine the flux at that particular coordinate. In other words, the mixing ratio $X_C$(s,z), air density $\rho_{air}$ (s,z), and normal wind speed $U_\perp$(s,z) are combined to arrive at a flux. Then, the fluxes at each location on the screen are integrated to obtain the total emission rate:

$$E_{C,H} = M_R \iint X_C \rho_{air} U_\perp ds dz$$

We have now added each equations of how to calculate each term emission rate ($E_{C,H}$, $E_{C,V}$, and $E_{C,M}$) in the main text in Section 2.5 Mass balance approaches for determining emission rates.

The assumption in TERRA is that during the flight time of 30 min, the wind field and hence the plume remains at a steady state. Since the flights created virtual screens through which the plume will be transported through, no assumption is needed for gas dispersion for the emission computation. Higher windspeed with lower mixing ratios and lower windspeed with higher mixing ratios for the same flux. Their integration over the virtual screen yields the emission rate, hence the magnitude of windspeed during the period of measurement does not affect the total emission rate.

Supplement - Line 84 - the extrapolation comparison is a good idea. I'd like to see the results reported. It probably doesn't make much difference in your case.

We have reported the emission results using different extrapolation techniques in Section 5.3 Uncertainty analysis

**Technical corrections**

Line 95 Supplement - "TERRA"

Corrected.

Line 101 - these should be given at least one more signifcant figure in my opinion, and differentiation should be made between measured and estimated values

Line 101 corresponds to **Table S1. Assessment of percent uncertainties for CH₄ and CO₂ emission estimations from the two coking plant stacks**, which listed each term of uncertainty and the total uncertainty. This table does not involve the difference between measured and estimated values.

Maybe you mean Line 110 which corresponds to **Section SI-3. Evaluation of CO₂ emissions through carbon material balance**? Then the below figure is made to present the difference between measured and estimated values. This graph has also been added to the SI.

[Figure]

Line 324 - "satellite"

**Last word**

I really enjoyed reading this paper, and what I have read is of high quality, and a really interesting and unusual application of methods that I haven't seen before in this combination. It will make a great paper - I hope these comments are helpful and not disheartening. Great work!

We really appreciate your comments, and we hope our replies to your comments clarified some ambiguities in the original draft, as well as answered your questions. The revised paper is indeed a better manuscript.

**Referee#2**

Review of "Application of a new UAV measurement methodology to the quantification of $CO_2$ and $CH_4$ emissions from a major coking plant" by Tianran Han et al.

This paper presents a new UAV-based system to measure $CO_2$ and $CH_4$ emissions from a local anthropogenic source. I think the paper addresses a relevant question and the presented UAV-based system is a novel combination of methods for quantifying GHG emissions. The paper thoroughly presents the measurement setup and data analysis methods including uncertainty estimation, and compares results from a test flight to different other methods. However, I did not find all the information where I expected it to be and I think the manuscript needs some revisions. I outline my general and specific comments below.

**General comments**

In many parts of the manuscript, I was missing additional information that I later found in the Supporting Information (SI). I think it would be helpful for the reader to have some of this information in the main manuscript or at least refer to the SI. This applies to some of my specific comments below.

Thank you for your suggestion. We have moved "**Section SI-1. TERRA-based determination of CH₄ and CO₂ emission rates**" and "**Section SI-2. Uncertainty estimation**" from Supporting Information to the main text.

Three pages of the manuscript (out of 15) cover the deconvolution of the measured mole fraction time series. I wonder if dedicating this amount of space is justified. Is the data deconvolution novel compared to previous UAV-based GHG measurements? In this case, its importance could be highlighted. Otherwise, which advantages does the deconvolution method offer compared to a simple response time correction? In contrast, the manuscript contains very little information about the other uncertainty sources and the actual emission quantification method.

Yes, data deconvolution is a novel technique used in UAV-based GHG measurements in that it restores the resolution of the mixing ratios from the tubing in the air sampler, issues that were not

solved previously. This is not the same as a simple response time correction; a response time correction merely shifts the data series in time but does not solve the issue of data smoothing inherent in the sampler and the resulting loss of time resolution. The importance of high time resolution in the measurement data allows for spatial resolution in the measured plume and allows for better data matching up with the meteorological data which are at 1-s time resolution.

To achieve a more balanced presentation, we have added information on uncertainty sources and emission quantification algorithms to the main text.

Speaking about the uncertainty sources, these are nicely listed in the SI, table S1. The deconvolution contributes only 1%, and there are other factors contributing at higher percentages. This could be discussed in the manuscript. Also, I was wondering if it would be feasible to measure additional flight legs at the top of the measurement volume box to reduce this uncertainty factor. (Would the wind speed measurements and/or the limited measurement time allow those measurements?)

Thanks for the suggestion! The uncertainty analysis has been moved to the main text. Also, we added more details to make the analysis clearer. Measuring additional flight legs at the top of the box will make a more accurate estimation of the vertical flux through the box top, and if the perimeter is not too large (less than 3km), the measurement time is enough for these additional flight legs. Actually, we have conducted a similar flight pattern previously (Graph below). During the measurement campaign at the coking plant in this study, we have a favorable meteorological condition to completely capture the plume on the lateral sides of the box, so flight legs at the top were not conducted.

[Figure]

I think it would be helpful to refine the structure of the manuscript. The methods section should contain information on how emissions are derived from the air sampler and wind measurements (instead of in paragraph l. 277ff). Novelties or improvements to established methods could be emphasized. Sections 3 and 4 could be merged since they both refer to the laboratory tests and subsequent corrections.

A sub-section has been added to the method section to provide information on the emission

quantification method (Section 2.5 Mass balance approaches for determining emission rates). The modifications to the TERRA algorithm were introduced in an individual paragraph as follows:

"To suit the UAV measurements, the following modifications to the TERRA algorithm were made: (1) A much higher interpolation resolution for the kriging mesh was implemented for application to the UAV measurements in this study, with the interpolation mesh size adjusted to 1 m (vertical) by 2 m (horizontal), as UAVs fly significantly shorter distances compared to applications to piloted aircraft for which the interpolation resolution was 20 m (vertical) by 40 m (horizontal); (2) The modified TERRA now applies an embedded routine to automatically fit flight tracks using least squares, while this procedure was previously conducted manually offline through geographic information system when using TERRA. (3) The modified version of TERRA has added an algorithm for correcting negative weights during Kriging interpolation following Deutsch (1995)."

Section 3 and 4 have been merged as follows:

3 Laboratory tests
3.1 Validation of the air sampler
3.2 Data deconvolution to achieve high time resolution

**Specific comments**
l.14: The first sentence and the subsequent lines in the abstract focus on measuring GHG, it might be good to emphasize the emission quantification right at the beginning.

We have changed this sentence to:

"The development in unmanned aerial vehicle (UAV) technologies over the past decade has led to a plethora of platforms that can potentially enable greenhouse gas emission quantification."

l.16: The 150m length information at first sounds confusing when mentioned together with the UAV. Maybe add "coiled".

"coiled" has been added.

l.18 "During flights, the air sampler starts sampling as soon as the UAV takes off, and stops sampling after landing." Is this relevant to the abstract?

This sentence has now been deleted.

l.75/76: Even though using different $CO_2$ measurement methods, $CO_2$ emissions have been quantified with a UAV-based system: https://doi.org/10.5194/amt-14-153-2021. Other studies describe measuring mere $CO_2$ concentrations with UAVs (https://amt.copernicus.org/articles/15/4431/2022/, https://www.mdpi.com/2073-4433/10/9/487, https://amt.copernicus.org/articles/16/513/2023/). Maybe those studies could be discussed?

Thanks for providing the information! We have added the following content:

"Studies of using UAVs for $CO_2$ plume detection and mapping from anthropogenic sources have also been reported (Reuter el al., 2021; Liu et al., 2022; Leitner et al., 2023; Chiba et al., 2019) Reuter et al. presented the development of a UAV platform to quantify the $CO_2$ emissions of anthropogenic point sources by deployment of an NDIR (non-dispersive infrared) detector and a 2-D ultrasonic acoustic resonance anemometer on the platform (Reuter et al., 2021)."

l.78/79: "three-dimensional measurements of $CO_2$ and $CH_4$" To me that sounds like a continuous space-time measurement. "on a trajectory in the three-dimensional space" might be more suitable.

This sentence has been revised to:

"In this study, we developed a new active air sampling system for deployment from a UAV on a trajectory in the three-dimensional space to measure $CO_2$ and $CH_4$"

l.88ff: Trying to research the Tier 1 method, I could find neither the "IPCC 2006" nor the "Ministry of Ecology and Environment of of China, 2018" citation in the reference list, but a "Kopp et al. 2021" reference (Physical Science basis of the IPCC 6th assessment report). It would help to point the reader to the specific report/chapter where the information can be found. Also, it might be worth mentioning that the Tier 1 method is the least complex method for estimating GHG emissions.

We have added the following references to the reference list:

"IPCC, 2006 IPCC Guidelines for National Greenhouse Gas Inventories, Chapter 4: Metal Industry Emissions."
"Ministry of Ecology and Environment of China, The People's Republic of China Second Biennial Update Report on Climate Change, Part II National Greenhouse Gas Inventory, Chapter 1.3 Industrial Process, 2018."

We have also added this sentence to the main text:

"Tier 1 methodologies are the simplest and least complex requiring less resources on collection the necessary data and producing GHG emission estimates."

l.95ff: It is stated that the new system is based on the AirCore system, which seems to be an established method for airborne $CO_2$ and $CH_4$ mole fractions. What has been changed or improved compared to the existing AirCore system?

The AirCore system contains a 150-m-long stainless steel tube, open at one end and closed at the other, that relies on positive changes in ambient pressure for passive sampling of the atmosphere. (Karion, 2010) . We have now added this brief introduction to the existing Aircore system in main text.

The new air sampling system in this study encompasses a pump, a micro-orifice, a $CO_2$ marker generator, two three-way solenoid valves, and electric relays in addition to the tubing, allowing for active automatic sampling

l.119ff: Which parameters measured by the sonic anemometer are used? Only mean wind speed and direction? Vertical and/or horizontal wind components? Which resolution and why? Is the measured temperature the sonic temperature (as measured by standard sonic anemometers) or the actual air temperature? The text repeatedly refers to "meteorological parameters", but it is not explained which are used.

The anemometer measures three-component of wind speed $(U_x; U_y; \omega)$ and temperature (T) at 3 Hz. These four parameters are all used for TERRA calculation. Temperatures are used to calculate air density and the rate of pollutant mass added to or subtracted from the total box due to changes in air density with time. The measured temperature is the sonic temperature.

The three-component wind speed was further transformed into wind speed and wind direction after corrections for attitude (pitch, yaw, roll) changes, the UAV airspeed, the perturbations caused by the UAV rotor propellers using a correction algorithm developed by our group (Yang et al., paper ready for submission and is uploaded for the review of this manuscript). The final wind data were paired to the $CO_2/CH_4$ data, and each wind/concentration data pair were used in the TERRA computation. Wind data were not averaged.

We have added the following content to the main text:

"The anemometer measures three-component wind speed $(U_x; U_y; \omega)$ and temperature (T). The measured data were further transformed into actual wind speeds and wind directions after corrections for UAV attitude (pitch, yaw, roll) changes and accounting for its airspeed, as well as the perturbations caused by the UAV rotor propellers using a correction algorithm developed in our group (Yang et al., paper ready for submission)."

l.140ff: I also would like to hear more details about how the correction for rotor-induced air flows and UAV attitude and motion was performed. The Yang 2023 paper is not published, so no details are given about these corrections. Related to that, which instruments were used to determine the UAV airspeed and attitude, the GPS sensor (which?) mentioned in the section after?

More details of the post-flight correction for wind speed and direction have been added to the main text in. We have also uploaded the manuscript of Yang's work on wind speed correction for your reference.

The GPS information, airspeed, and attitude data (pitch, yaw, and roll) were extracted from the UAV data transmitted to the ground control station. We have added this information to the main text.

l.160 ff, Fig. 2, Fig. 3: It would be helpful to be more consistent with how you name different instruments/setups. Using different words to describe the same thing makes it hard to understand

the validation. (E.g. Picarro=online measurement= CRDS ="same analyzer"?, air sampler = presented system?, Artificial source = lab air = mix of standards of $CO_2$, $CH_4$?). Did I understand correctly that adding the Picarro in Fig2a is the validation reference, while the general process in Fig 2a and Fig. 2b shows the standard process of the air sampler? I also did not understand the purpose of the zero air in Fig 2b (it is mentioned in l. 197 for the first time).

We have revised this section to make the naming clearer and more consistent:

"Prior to flights in the field, we validated the air sampler in laboratory experiments by first sampling an artificial air while making simultaneous online measurements of the artificial air with the CRDS analyzer, and then analyzing the sampled artificial air was with the same CRDS analyzer and comparing the results from the air sampler to the online measurements. An experimental apparatus was constructed for the simultaneous sampling of the same artificial air with the air sampler and the CRDS analyzer through a tee junction (Fig. 2(a)), and subsequent air sample analysis using the same CRDS analyzer (Fig. 2(b)). In the artificial air, $CH_4$ and $CO_2$ standards were control-released into the lab air from an 8 L gas cylinder filled with a gas mixture of 5 ppm $CH_4$, 2 ppm CO and 600 ppm $CO_2$ to generate the artificial air source. The outlet of the standard gas cylinder was held at varying distances to the tee junction over time to yield a time series of different $CH_4$ and $CO_2$ mixing ratios, which was designed to mimic plumes expected in the real atmosphere. During analysis, the flow rate through the zero air (Mass Flow Controller 1) is adjusted to make sure that the flow rate through the air sampler (Mass Flow Controller 2) is stable and consistent at 54 sccm (Section 2.4).

Figure 3 (a) illustrates the mixing ratios of $CO_2$ and $CH_4$ time series obtained from the air sampler and online measurements by the CRDS analyzer. It can be seen that the measured results from the air sampler and the CRDS analyzer are in good agreement throughout the tests, and the correlation coefficient is estimated to be 0.89 and 0.73 for $CH_4$ and $CO_2$ (Figure 4(c) and (f)). For the measurements with the air sampler, short term variations and noises in the $CH_4$ and $CO_2$ mixing ratios, that were fully captured by the CRDS analyzer during the online measurements, were smoothed out, while the main features and tendencies were preserved. In fact, the air sampler measurement results should be a smoothed version of the CRDS analyzer online measurements, due to mixing in the analyzer cavity, molecular diffusion during sample storage in the sampler, inner wall surface drag and desorption during its withdrawal from the tubing during analysis, as well as Taylor dispersion during sampling and analysis (Karion et al., 2010). Dilution with zero air during later CRDS analysis also contributes to the smoothing."

In Fig. 2a, Picarro is the validation reference. Fig. 2b is the standard process for analyzing the samples stored in the air sampler. Zero air in Fig2b is used for dilution purposes because the flow rate of the pump in Picarro is much larger than that in the air sampler.

The zero air was mentioned for the first time in Line 158 in Section 2.4 Air Sample Analysis. The zero air mentioned in Line 197 is for creating the one-second concentration pulse signal used to obtain the convolution kernel.

l.175: I think "good agreement" could be supported by a correlation plot or quantifying a correlation

coefficient and errors. These numbers might also show if/how the correlation improves for the convoluted signal introduced later.

Fig. 3: Also related to the naming of the signals, it could be helpful to mark which lines belong to which original signal and be consistent with assigning labels. Additionally, it might help to include a correlation plot instead of the current Fig. 3b and make Fig. 3b a separate figure.

We have added the correlation plots to Fig 3 and made Fig.3b a separate figure:

[Figure]

We also changed the color of the signals in the graph above so that the signals of the same color represent the original signals and the corresponding signals after convolution or deconvolution.

Corresponding editings have also been made to the main text:
"It can be seen that the measured results from the air sampler and the online CRDS measurements analyzer are in good agreement throughout the tests, and the correlation coefficient is estimated to be 0.89 and 0.73 for $CH_4$ and $CO_2$ (Figure 4(c) and (f))"

"The deconvolved series of $CH_4$ and $CO_2$ restored with the Wiener convolution filter are shown in Fig. 4(a) and (b), and the correlation coefficient between the deconvoluted results and the online measurements with the CRDS analyzer are 0.93 and 0.79 for $CH_4$ and $CO_2$ (Fig. 4 (e) and (h)), higher than those between the original air sampler measurement and the CRDS analyzer."

"To test whether the kernel weights $\hat{g}(t)$ can smooth the online measured concentrations from the first lab experiment (top line in Fig. 4(a) and (b)), the weights $\hat{g}(t)$ were used to convolute with the data from the online measurements (i.e., $x(t)$), resulting in an estimated $\hat{y}(t)$ (Fig. 4(a) and (b), third line) that is in excellent agreement with the measurements from the air sampler, with the correlation coefficients increased to 0.99 and 0.98 for $CH_4$ and $CO_2$ (Figure 4 (d) and (g))"

l.183ff: Is the deconvolution a novelty added to the original AirCore sampler? In the section above, the Karion et al., 2010 study is cited listing different error sources. Were the error sources quantified and/or mitigated in the original AirCore methodology?

Yes, deconvolution is a novelty that hasn't been done before. These error sources were evaluated in laboratory tests, and they were quantified by a model of molecular diffusion and flow-induced mixing in the original AirCore methodology, which is a different method from the method in this study.

Fig. 3b: Please label the axes more clearly (unit of time on the x-axis, normalized concentration on the y-axis?). The two plots could also be merged into one with two curves, where the colors match Fig. 3a.

Thanks for the suggestion! The plot has been edited:

[Figure]

l.274: It seems like the elevation of the coking plant is nearly sea level, but it might make more sense to indicate flight altitude in AGL.

We have edited to indicate flight altitude in AGL:

"The UAV ascended from the ground to 135 m a.g.l. and started the box flight at this altitude, ascending 15 m every level and reaching a maximum altitude of 255 m a.g.l."

l.277: This paragraph (maybe better located in the methods section) needs further explanation. Which input parameters are used (specify "meteorological data")? The control volume is defined by the flight path of the UAV? Are all terms in the mass balance (Table 2 in Gordon et al., 2015) considered? Why does the UAV not measure in the plane of the top of the box (which is later on listed as an uncertainty)? In the SI, it is explained that the vertical transport through the top of the box is neglected, but it might become important under unstable conditions, so information about atmospheric stability during the measurements would be interesting.

We have moved the introduction of the TERRA methodology to the Section 2.5 Mass balance approaches for determining emission rates and deleted the original paragraph in Line 277. The present plumes were completely captured on the downwind virtual screen, so that vertical transport in the mass balance computation was insignificant and negligible. Section 2.5 now provides detailed information on how TERRA calculation is done. The meteorological data were specified as follows:

"The UAV-based measurements were coupled with the mass-balance approach TERRA to determine the emission rates of the measured pollutants using their measured mixing ratios and the meteorological data (three-component wind speed ($U_x$; $U_y$; $\omega$) and temperature (T)) collected on board the UAV during the flight"

The control volume is defined by the flight path of the UAV, this is explained in the added Section 2.5:

"To run TERRA based on a virtual box flight, the first step is to map the $CH_4$ and $CO_2$ mixing ratio data measured along the level flight tracks encircling a facility to the two-dimensional virtual walls of the virtual box,"

Only $E_{C,H}$, $E_{C,M}$, and $E_{C,V}$ were considered in the study, as other terms contribute very little to the total emission rates (Gordon et al., 2015). We have added this information to the text as follows:

"Other terms listed in the Gordon et al. computation algorithm that were used to solve for the total emission rate were often neglected as they contribute little to the total emission rates (Gordon et al., 2015)."

During the measurement campaign at the coking plant in this study, we had a consistent wind speed to completely capture the plume on the lateral sides of the box, so we did not conduct the measurement in the plane of the top of the box.

Actually, the vertical transport through the top of the box was not neglected, it is considered as $E_{C,V}$, We were trying to explain that the vertical term is normally neglected in other top-down approached. Perhaps the wording has led to misunderstanding. The text has been revised as follows:

"The vertical transfer rate term $E_{C,V}$ is estimated by computing the air mass vertical transfer rate, determined from vertical wind estimated from air mass balance within the box, and multiplying it with the $CO_2$ or $CH_4$ mixing ratios at the box top. This term is normally negligible in other top-down emission estimate approaches since it is typically miniscule compared to horizontal fluxes, but can affect the computed emission rates when vertical air movement becomes more significant such as under unstable atmospheric conditions."

l.284: How is the original TERRA further modified and tailored?

The modifications to the TERRA algorithm were introduced in an individual paragraph as follows:

"To suit the UAV measurements, the following modifications to the TERRA algorithm were made: (1) A much higher interpolation resolution for the kriging mesh was implemented for application to the UAV measurements in this study, with the interpolation mesh size adjusted to 1 m (vertical) by 2 m (horizontal), as UAVs fly significantly shorter distances compared to applications to piloted aircraft for which the interpolation resolution was 20 m (vertical) by 40 m (horizontal); (2) The

modified TERRA now applies an embedded routine to automatically fit flight tracks using least squares, while this procedure was previously conducted manually offline through geographic information system when using TERRA. (3) The modified version of TERRA has added an algorithm for correcting negative weights during Kriging interpolation following Deutsch (1995)."

l.297: What are the $CO_2$ markers? They are shown in Fig. 4, not Fig.5.

The air samples in the tubing are analyzed with the Picarro CRDS analyzer after UAV flight, thus we need to know when the sampling begins to identify the starting point of the time series. The $CO_2$ markers result from pulses of air with low/zero $CO_2$ concentrations, producing low deviations in the Picarro instrument output from the signals for the background air $CO_2$ concentrations. $CO_2$ markers are set at the beginning of every flight and at specifically set times, so that they produce very evident negative signals in the $CO_2$ time series. These markers allow us to match the starting time of the sampling to the data point of the first $CO_2$ marker and later the other times when the markers are known to be set. We have added the following sentence to the main text:

"The $CO_2$ markers help to identify the starting point and specific times subsequently during the UAV air sampling in data extraction and analysis."

l.321: "The uncertainties for the estimates were derived from detailed analyses of each uncertainty source": Here, I would like to hear more details about how the different uncertainty sources contribute to the overall uncertainty.

We have added more calculation process and details for deriving each term of uncertainty in Section 5.3 Uncertainty analysis:

"The accuracy of the mixing ratio measurements from the Picarro CRDS analyzer is 50 ppb and 1 ppb for $CO_2$ and $CH_4$, respectively. By adding variations in the measured mixing ratios based on the measurement accuracies and re-applying TERRA, the derived emission rates varied within 1% for both $CO_2$ and $CH_4$. Thus, the uncertainties in the emission rates due to mixing ratio measurements ($\delta_M$) were estimated at 1% for both $CH_4$ and $CO_2$.

The anemometer measures wind speeds with an accuracy of $\pm 0.1$ m s$^{-1}$ at wind speeds <10 m/s and wind directions with an accuracy of $\pm 1°$. The uncertainty of the wind measurements ($\delta_{Wind}$) was estimated using error propagation in the normal wind $U_\perp$(s,z), as it is calculated from the northerly and easterly wind components, thus from wind speed and wind direction:

$$\delta_{U_\perp} = \sqrt{\delta_{easterly}^2 + \delta_{northly}^2 + 2\sigma_{easterly-northly}}$$

$$\delta_{easterly} = |WScos(WD)\sigma_{WD}|$$
$$\delta_{northly} = |WSsin(WD)\sigma_{WD}|$$

[revised manuscript text omitted]

l.387 – 389: The last sentence might be a bit too general since only one measurement case is studied. Also, your results show that the IPCC emission factor was too high only for $CO_2$, not for $CH_4$.

We have revised the sentence into:

"In addition, when compared the top-down UAV-based measurement results to that derived from the bottom-up emission inventory method, the present findings indicated that the IPCC emission factors can be significantly different from the actual emission factors."

**Technical corrections**
I could not find some citations in the reference list (e.g., IPCC 2006; Ministry of Ecology and Environment of China, 2018; European IPPC Bureau, 2001). Please make sure that the reference list is complete.
Sorry for missing the reference. They have been added to the list now

"European IPPC Bureau, Integrated Pollution Prevention and Control (IPPC) Best Available Techniques Reference Document on the Production of Iron and Steel, 2001."
"IPCC, 2006 IPCC Guidelines for National Greenhouse Gas Inventories, Chapter 4: Metal Industry Emissions."
"Ministry of Ecology and Environment of China, The People's Republic of China Second Biennial Update Report on Climate Change, Part II National Greenhouse Gas Inventory, Chapter 1.3 Industrial Process, 2018."

Please make sure to spell proper names correctly (AirCore, Gålfalk, TriSonica).

Corrected

The authors might consider using "uncrewed aerial vehicle" instead of "unmanned aerial vehicle" if this is not a firm convention.

Corrected

It might be helpful to use consistent wording, e.g. do "mole fraction" and "mixing ratio" mean the same? Is "air sampler" the same as "Aircoil"?

Thanks! We have changed "mole fration" into "mixing ratio", and "Aircoil" into "the air sampler"

l.21: Please spell out CRDS when used first in the abstract.

Corrected

Fig. 1: Pump instead of bump?

Corrected

l.123: Is Geotech the manufacturer or just a distributor?

Geotech is the manufacturer

l.168: Please spell out MFC.

Corrected

l.292: Deconvolution is explained in Section 4, not 3 (if sections remain unchanged).

Corrected